# Effects of innovative long-term soil and crop management on topsoil properties of a Mediterranean soil based on detailed water retention curves

Alaitz Aldaz-Lusarreta[1,2], Rafael Giménez[1,2], Miguel A. Campo-Bescós[1,2], Luis M. Arregui[1,4], Iñigo Virto[1,3]

[1]Institute for Innovation & Sustainable Development in Food Chain (IS-FOOD), Universidad Pública de Navarra (UPNA), Campus de Arrosadia, 31006 Pamplona, Spain.
[2]Department of Engineering, Ed. Los Olivos, Universidad Pública de Navarra (UPNA), Campus de Arrosadia, 31006 Pamplona, Spain.
[3]Department of Science, Ed. Los Olivos, Universidad Pública de Navarra (UPNA), Campus de Arrosadia, 31006 Pamplona, Spain.
[4]Department of Agricultural Engineering, Biotechnology and Food, Ed. Los Olivos, Universidad Pública de Navarra (UPNA), Campus de Arrosadia, 31006 Pamplona, Spain.

*Correspondence to*: Alaitz Aldaz-Lusarreta (alaitz.aldaz@unavarra.es).

**Abstract.** The effectiveness of cconservation agriculture (CA), and other soil management strategies implying a reduction of tillage has been shown to be site-dependent (crop, clime and soil), and then any new soil and crop management should be rigorously evaluated before its implementation. Moreover, farmers are normally reluctant to abandon conventional practices if this means putting their production at risk. This study evaluates an innovative soil and crop management (including no-tillage, cover crops and organic amendments) as an alternative to conventional management for rainfed cereal cropping in a calcareous soil in a semi-arid Mediterranean climatic zone of Navarra (Spain), based on the analysis of soil water retention curves (SWRC) and soil structure. The study was carried out in a small agricultural area in the municipality of Garinoaian (Navarre, Spain) devoted to rainfed cereal cropping. No other agricultural area in the whole region of Navarre exists where soil and crop management as proposed herein is practiced. Climate is temperate Mediterranean and the dominant soil is *Fluventic Haploxerep*. Within the study area there is a subarea devoted to the proposed soil and crop management (OPM treatment), while there is another subarea where the soil and crop management is the conventional in the zone (CM treatment). OPM includes no-tillage (18 years continuous) after conventional tillage, crop rotation, use of cover crops and occasional application of organic amendments. CM involves continuous conventional tillage (chisel plow), mineral fertilization, no cover crops and a lower diversity of crops in the rotation. Undisturbed soil samples from the topsoil and disturbed samples from the tilled layer were collected for both systems. The undisturbed samples were used to obtain the detailed SWRCs in low suction range using a Hyprop© device. From the SWRCs, different approaches found in the literature to evaluate soil physical quality were calculated. The pore-size distribution was also estimated from the SWRCs. Disturbed samples were used in the laboratory to assess soil structure by means of an aggregate-size fractionation, and to perform complementary analysis from which other indicators related to soil functioning and agricultural sustainability were obtained. The approaches evaluated did not show

clear differences between treatments. However, the differences in soil quality between the two forms of management were better observed in the pore size-distributions, and by the analysis of the size-distribution and stability of soil aggregates. There was an overabundance of macropores under CM, while the amount of mesopores (available water) and micropores were similar in both treatments. Likewise, more stable macroaggregates were observed in OPM than in CM, as well as more organic C storage, greater microbial activity, and biomass. The proposed management system is providing good results regarding soil physical quality and contributing also to the enhancement of biodiversity, and to the improvement in water use efficiency. Finally, our findings suggest that the adoption of the proposed practice would not result in a loss in yields compared to conventional management.

## 1 Introduction

Conservation agriculture (CA), and other soil management strategies implying a reduction of tillage have been reported to reduce soil degradation –preserving soil structure and associated porosity– in different agroecological situations (Verhulst et al., 2010; Sartori et al., 2022), and in many cases are indeed designed for this purpose (Virto et al., 2015).

The reasons reported for its adoption in Europe are several. In Northern Europe soil erosion control, soil crusting in loamy soils and the need to increase soil organic C storage, as well as soil trafficability are widely cited as reasons for CA implementation (Lahmar et al., 2007). In the Mediterranean countries, soil water storage and water-use efficiency can be added to this list of reasons (De Tourdonnet et al., 2007). The most widely reported benefits of CA in Southwestern Europe in relation to erosion are the increased soil infiltrability and/or the protective effect of crop residues on the soil surface (Gómez et al., 2009; Espejo-Pérez et al., 2013; Virto et al., 2015). In Spain, the soil water-retention capacity has been observed to be greater in semi-arid land under no-tillage (Fernández-Ugalde et al., 2009; Bescansa et al., 2006). Other positive effects of CA on soil quality observed in semi-arid rainfed agricultural systems in Spain are related to soil organic C and nutrients storage (Ordóñez Fernández et al., 2007).

However, different studies show that the effectiveness of CA in solving these problems can be site-dependent (Costantini et al., 2020; Chenu et al., 2019), and variable depending on its effect on crop yields (Virto et al., 2012).

Indeed, since crop performance under no-till is strongly dependent on the crop type and climate (Or et al., 2021) and also soil type, no-till may not be suitable for all conditions (Pittelkow et al., 2015). In fact, in some areas, no-till often results in reduction in crop yields of ca. 10% (Or et al., 2021). Conventional tillage –in carefully managed agricultural soils– may be imposed when no-tillage would lead to chronic and unacceptable yield losses.

From the perspective of the effects of CA on the soil, among the existing approaches to assess soil condition (Minasny and McBratney, 2018), Rabot et al. (2018) highlighted the interest of soil structure as an indicator of its performance, as well as the relevance of considering the organization, distribution, and stability of aggregates and the characterization of the associated pore system.

There are different types of techniques to characterize the soil pore system (Pires et al., 2013; Taina et al., 2013; Pagliai et al., 2004). The analysis of soil water retention curves (SWRCs) –the relationship between soil water matric potential and soil water content– is one of the most employed methods for characterizing soil pores. It enables an adequate characterization of the effective porous system (interconnected, functional pores) and therefore, SWRCs are a valuable tool to diagnose the physical condition of soils (Dexter, 2004a, b; Pires et al., 2017). In addition, it is a relatively fast and low-cost methodology.

One relevant issue in the assessment of the effects of soil management on soils is the increasing need to co-learn with farmers and other stakeholders (Bouma, 2014), and to identify the consequences of changes in land use in actual field conditions. Likewise, this assessment needs to account for as much soil functions as possible (Bünemann et al., 2018), as recently suggested from the perspective of linking soils with sustainable development goals (SDG, (Lal et al., 2021; Bouma et al., 2021)).

In this framework, the objective of this study was to assess the continuous application, throughout 18 years, of an innovative soil and crop management –in comparison with conventional management– for the improvement of the soil physical condition, and the optimization of the soil water balance, in rainfed cereal agrosystems in semi-arid land (Navarre, Spain). It has to be emphasized that there is –to our knowledge– no other agricultural area in the whole region of Navarre where soil and crop management as proposed herein is practiced –and even less for almost two decades– with the exception of precisely our test area.

Base on the analysis of detailed SWRCs and soil structure, (i.e., the size-distribution of stable macro- and microaggregates), and its consequences in soil water retention, the evaluation includes other complementary aspects relevant to soil functioning and SDGs by assessing soil organic C storage (climate regulation, SDG #13), the soil biological diversity (biodiversity loss, SDG #15) and (as far as available from farmers) yields (food security, SDG #2). This evaluation aims to incorporate therefore real-case field-measured indicators, in line with the recent recommendations of the new European Agricultural Policy (Bouma et al., 2022; Panagos et al., 2022).

## 2. Material and methods

### 2.1. Study zone and treatments

The study was carried out in a small agricultural area in the municipality of Garinoaian (Navarre; 42,59843° N, 1,64959° O). This is an area with a Csb type of climate according to the Koppen-Geiger classification (Gobierno de Navarra Meteorología y Climatología de Navarra, 2022; Peel et al., 2007) The mean annual reference evapotranspiration according to the FAO Penman-Monteith method is 1107 mm·year$^{-1}$. For crops in the rotation, the mean annual crop evapotranspiration is 326 mm·year$^{-1}$. The soil –*Fluventic Haploxerepts* (Soil Survey Staff, 2014), *Fluvic Cambisol,*(FAO, 2015) – is devoted to rainfed

cereal cropping. The physical-chemical properties of the soil (Table 1) showed high homogeneity of the material at the study depth (0-30 cm) regarding the most relevant physical-chemical properties related to moisture retention, except for the content of organic C (which can be related to the change in management). In addition, *in situ* standard soil description corroborated the homogeneity of the topsoil (0-30 cm).

The physical-chemical properties of the soils shown in Table 1 was done using standard methods. In particular, soil pH was

100 analyzed in a 1:2.5 soil:water solution as in Hendershot and Lalande (1993), organic C content by wet combustion as in Tiessen and Moir, (1993), carbonates were determined in a modified Bernard's calcimeter following Pansu and Gautheyrou (2003a), and the electrical conductivity in a soil:water solution similar to that for pH analysis (Pansu and Gautheyrou, 2003b). The soil texture was determined by the pipette method. All analyses were conducted on air-dried samples ground to 2 mm, collected at 0-30 cm, as proposed, for example, by FAO for organic C storage (FAO, 2020). Finally, the bulk density was determined using

the Hyprop© device (see section 2.2) from undisturbed samples extracted from the first 5 cm of the soil profile. However, based on field standard soil description, it is fairly safe to assume that the bulk density is roughly constant up to 30 cm depth.

**Table 1. Physical-chemical properties of the soil (0-30 cm) in OPM and CM treatments and the textural characterization of both treatments. Mean ± standard deviation of the mean (n=3). Statistically significant differences (p < 0.05) are in bold.**

Within the study area there is a subarea –to our knowledge, unique in Navarre– devoted to a pioneer optimized soil and crop management (from now on OPM treatment). There is another subarea –adjacent to OPM one– where the soil and crop management is the conventional in the zone (from now on CM treatment).

OPM is an optimized system, used for 18 consecutive years, which includes direct seeding, an improved crop rotation including wheat (*Triticum aestivum L.*), barley (*Hordeum vulgare L.*), legumes (*Pisum sativum L.*, *Vicia faba L.* and others) and rapeseed

(*Brassica napus L.),* and the occasional use of cover crops and organic amendments. Both grain and straw were removed in the 11 first years of implementation, and only stubble remained on the surface of soil when direct seeding was implemented with minimal soil perturbation. Since then, and for the 7 remaining years, the procedure was slightly modified, and only grain

was removed at harvest. Therefore, chopped straw and stubble remained on the surface of the soil before direct seeding with no disruption of the soil surface. At the same time, cover crops were introduced in the system, despite this being a risky practice in rainfed Mediterranean agrosystem characterized by warm and dry summers. As such, summer cover was routinely granted in this system by letting spontaneous vegetation grow in the summer, after harvest. This vegetation was controlled with herbicides before seeding the cash crops in the fall. Also, only one year the winter crop used was *Vicia villosa* Roth, and served as a cover crop for sorghum (*Sorghum vulgare* L.), which was successfully grown in the spring-fall season despite the limiting water availability in the area.

CM is a conventional management, which employs conventional continuous (annual) tillage with a chisel plough down to 15 cm, mineral fertilization, without cover crops, and a less diverse crop rotation including mostly wheat and occasionally legumes and rapeseed. Crop residues are not returned into the soil (both grain and straw were removed annually): only the non-exported stubble and roots were therefore incorporated into the soil at 10-15 cm depth by vertical tillage.

In both treatments (OPM and CM), mineral fertilization consisted of phosphorus addition before seeding (120-150 kg·ha$^{-1}$ of triple superphosphate 0-46-0) and nitrogen supply of 180 kg N·ha$^{-1}$ (split and distributed into two cover dressings at 60 kg N·ha$^{-1}$ and 120 kg N·ha$^{-1}$ in January and March, respectively) as urea. Organic fertilization was not used in any of the study treatments until 2021, in which an organic amendment was applied to the soil without disturbing the surface in the OPM treatment. After harvest, pig slurry was applied with an average concentration of 2.5 kg N·m$^{-3}$, by means of a tanker equipped with a system of hanging pipes that deposit the product a few centimeters above the ground and at a time close to a forecasted rainfall event. The application rate was 60 m$^3$·ha$^{-1}$ of slurry. These rates are within the legal limits established by legislation for groundwater protection against pollution caused by nitrates from agricultural sources (EU Directive 91/676 (Council of the European Union, 2008)), as the area is within a vulnerable watershed according to this Directive.

To avoid the possible influence of the preceding crop, it was ensured that the two last crops of the rotation before the study both in OPM and CM were the same (winter wheat (*Triticum aestivum L.)* and rapeseed (*Brassica napus L.*)).

**2.2 Soil sampling and methodological approach**

Soil sampling for both treatments (OPM and CM) was carried out in early fall –after harvest and before soil preparation for seeding in CM, approximately four months after the last tillage for CM– at three (n=3) randomly selected sampling sites per treatment: undisturbed cylindrical (8 cm diameter, 5 cm height) samples were collected from the first 5 cm of each sampling site.

In addition, in the same points, 3 disturbed composite samples –comprising 3 subsamples each– were taken at 0-30 cm depth for further physical-chemical and biological analysis in the lab. Immediately after sampling, part of the composite soil was stored at 5 ºC for biological analysis, while the remainder was used to assess soil aggregation, as detailed below.

*Determination of SWRCs.* From the undisturbed cylindrical samples, SWRC tracks were obtained in the laboratory with a Hyprop© device commercialized by METER (München, Germany) as described by Schindler et al. (2010). This device uses the Peters and Durner (2008) and Schindler (1980) simplified evaporation method. The procedure is based on the continuous measuring of matric component of soil water potential from two micro-tensiometers inserted into the saturated soil sample, while the moisture content of the sample is progressively reduced by evaporation. As the experiment advances, the sample loses water by evaporation, and the tensiometers record the variation of suction as a scale measures the weight change. The registries of suction and weight are automated and continuous. Gravimetric water content can be expressed as volumetric content since bulk density is known (Schindler et al., 2010).

After the evaporation experiment concluded, the samples were dried in an oven at 105 ºC for 24 h to determine the dry weight and the soil bulk density, for the subsequent evaluation of the results using the Hyprop-Fit (version: 4.2.2.0) software (Pertassek et al., 2015). In total, around 100 evenly distributed suction-water content between 0 kPa and 150 kPa were measured; with an extra measurement at 1500 kPa (classical wilting point) obtained using a pressure plate (Dirksen, 1999). The classical concept of permanent wilting point at a suction of 1500 kPa facilitates comparisons since it is widely used in the literature, though it should be taken with caution since it is not a universal wilting limit. Wiecheteck et al.'s (2020) findings when comparing the classical permanent wilting limit with the biological wilting of wheat and barley suggest that wilting depends on soil texture, with an occurrence of wilting at lower suction (i.e., wetter soil conditions) for sandy soils than for clay soils.

### 2.3 Analysis of the SWRCs and derived indices and functions

First, it should be noted that different mathematical functions to adjust SWRCs are found in the literature depending on the general shape of the SWRC. The SWRC of most soils presents a J form, defined by the presence of the air-entry region, in which the volumetric water content is maintained at saturation values even in suctions slightly over zero; this occurs due to occluded pores (not functional) (Kosugi et al., 2002). Instead, when there is no marked air-entry region, the SWRC adopts an S form. For instance, in the case of fine-textured undisturbed soils, the SWRC usually presents the shape of an S (Kosugi et al., 2002). Following Brooks and Corey (1964), in J-shaped SWRC the best fit occurs with an exponential function. But, for S-shaped SWRC, the fit with exponential functions is poor (Milly, 1987; van Genuchten and Nielsen, 1985), and it is recommended to employ sigmoidal-type functions such as the van Genuchten Equation (1980).

*Predicting soil water retention by uni-modal approaches. S index.* Dexter (2004a) proposed an S index to estimate the physical condition of soils (changes in soil structure, and therefore in porosity) based on the soil SWRC. This index represents the value of the slope of the SWRC at the inflection point when the curve is expressed as the natural logarithm of suction (in hPa) versus the gravimetric moisture content, $\theta g$ (kg·kg$^{-1}$) (Dexter, 2004a, b). According to Dexter (2004a), this inflection point defines the limit between structural pores (in the range of low suction) and textural pores (in the range of high suction values). It is

assumed that, as S increases, structural pores are more abundant and, therefore, there are better conditions for water flow and storage in the soil (Dexter, 2004a).

The inflection point can be determined directly by hand from the SWRC if there are enough accurate measurement points (Dexter, 2004a). Alternatively, it would be more appropriate to fit the SWRC to a mathematical function and then to calculate the slope at the inflection point in terms of the parameters of the function. To do this, one of the best-known functions is that proposed by van Genuchten (1980) for which, in turn, pedo-transfer functions are available for estimation of its parameters (Dexter, 2004a).

The value of S was calculated in two different ways assuming a uni-modal pore size distribution: i) from a sigmoidal function fitted to experimental data (Eq. 1), and ii) from the adjusted parameters of the van Genuchten (1980) function (Dexter, 2004a) (Eq. 2). To this end, the whole dataset was used, i.e., 0-150 kPa and 1500 kPa.

$$y = \frac{a}{1 + e^{-(\frac{x - x_0}{b})}} \tag{1}$$

Where $y$ is the logarithm of suction (hPa), $x$ is the gravimetric moisture (kg·kg$^{-1}$), and $a$, $b$, $x_0$ are parameters of the equation.

$$\theta_h = (\theta_{sat} - \theta_{res})[1 + (\alpha h)^n]^{-m} + \theta_{res} \tag{2}$$

Where $h$ is the soil matric potential (hPa), $\theta_h$ (m$^3$·m$^{-3}$) is the measured soil water content at matric potential $h$, $\theta_{res}$ is the residual water content (m$^3$·m$^{-3}$), $\theta_{sat}$ is the saturated water content (m$^3$·m$^{-3}$), $\alpha$ (hPa$^{-1}$), $n$ (-) and $m = 1 - (1/n)$ (-) are the van Genuchten parameters.

*Predicting soil water retention by a bi-modal approach*. Likewise, the water retention data was fitted to the double-exponential

equation with 5 adjustable terms proposed by Dexter et al. (2008), in which all the parameters have a different physical meaning (Eq. 3). To this end, the dataset between 0-150 kPa was used.

According to Jensen et al. (2019), this model can reflect better the effects of management systems in the soil properties.

$$\theta = C + A_1 e^{(-\frac{h}{h_1})} + A_2 e^{(-\frac{h}{h_2})} \tag{3}$$

Where $\Theta$ is the gravimetric water content; $C$ is the residual water content (asymptote of the equation); the amount of matrix

and structural pore space are proportional to $A_1$ and $A_2$, respectively. The values of $h_1$ and $h_2$ are the characteristic pore water suctions at which the matrix and structural pore spaces empty, respectively (Dexter et al., 2008).

*Numerical integration of SWRCs. Water retention energy index.* The water retention energy index (WRa) (Armindo and Wendroth, 2016) (Eq. 4) obtained from numerical integration including of each SWRC was determined.

$$WR_a = \int_{\theta_{pwp}}^{\theta_{fc}} h(\theta)\, d\vartheta \tag{4}$$

Where $\Theta_{fc}$ and $\Theta_{pwp}$ is the volumetric water content at field capacity and permanent wilting point, respectively; $h$ is suction (kPa).

WRa quantifies the total absolute energy that has to be applied by the soil to hold water in its pores between field capacity ($\Theta_{fc}$) –i.e., after the water drainage process becomes negligible– and wilting point ( $\Theta_{pwp}$) or any moisture point $\Theta_j$, where $\Theta_{pwp} \leq \Theta_j < \Theta_{fc}$. The WRa index was determined for the suction range between field capacity (ca. 10 kPa, see below) and a

moisture content corresponding to ca.150 kPa (maximum operating value of the Hyprop© device) which means a dataset of around 100 measured points (see above). It is clear that the accuracy of this index is highly conditioned by the degree of detail of the SWRCs.

This index presents an adequate sensitivity for smaller-scale, high-precision applications and for capturing the dynamic evolution of the soil physical state (Armindo and Wendroth, 2016). More precisely, in the case of two SWRCs measured before

and after some natural or anthropogenic changes (e.g., tillage), these energy indices can be used to quantify the change in soil physical quality status (Armindo and Wendroth, 2016).

*Estimation of field capacity.* The Hyprop© device, besides determining SWRC, provides values for soil unsaturated hydraulic conductivity at different water contents. From this, it is possible to estimate the moisture content of the soil at field capacity – i.e., once gravitational water is drained–.

*Estimation of pores size-distribution.* The soil pores size-distribution was estimated from the equivalent radius obtained from the suction values of SWRCs, using the equation formulated by Young and Laplace (Warrick, 2003) (Eq. 5):

$$h = \frac{2\, T \cos\theta}{\rho\, g\, r} \tag{5}$$

where $h$ is the height of the liquid (m), $T$ is the surface tension (N·m$^{-1}$), $\theta$ is the contact angle of the liquid, $\rho$ is the density of the liquid (kg·m$^{-3}$), $g$ is the gravitational acceleration (m·s$^{-2}$), and $r$ is the equivalent radius of the pores (m) retaining water at

a suction equivalent to $h$ (m).

## 2.4 Indicators of soil structure

*Aggregates size-fractionation*. Firstly, field-moist soil samples were gently passed through a 5 mm sieve, without forcing the aggregates, and left to dry naturally. Then 50 g were collected from each soil sample and subjected to humidification with deionized water vapor at room temperature, until saturation.

Water-stable aggregates fractionation followed the step-wise protocol described by Oliveira et al. (2019), as follows. Firstly, each moist soil sample was sequentially sieved (250 μm and 50 μm) to obtain three aggregate fraction sizes (Elliott, 1986) macroaggregates (Magg, > 250 μm), microaggregates (magg, 50-250 μm) and the silt and clay fraction ((s+c), < 50 μm). To this end, initially, 50 g of saturated soil sample were spread over a 250 μm sieve. The soil was then submerged in deionized water for approximately 30 seconds, and then manually sieved by moving the sieve upwards and downwards 15 times in a

distance of 1.5 cm during 30 seconds. The sieved material was then placed on a 50 μm sieve, submerged again for 30 seconds in deionized water, and the manual sifting was repeated. The sieved material was then transferred to a 500 mL centrifuge bottle, and centrifuged at x 13000 $g$ for 10 minutes to recover the silt and clay fraction. The aggregates retained by the sieves (> 250 μm and 50-250 μm) were gathered and dried in an oven at 50 ºC along with the fraction < 50 μm, and stored at ambient temperature for subsequent analysis.

The second step consisted in the fractionation of the > 250 μm fraction (Magg) in other three new fractions: coarse particulate organic matter > 250 μm (cPOM + sand), micro-aggregates within macroaggregates (mMagg, 50-250 μm) and particles < 50 μm within macroaggregates (M(s+c)). To this end, an *ad hoc* device adapted from Six et al. (2002), which consists of a block formed by a 250 μm sieve located above a 50 μm sieve, was employed. This block was placed on an agitator. Ten g of Magg (> 250 μm) and 50 glass beads (4 mm in diameter) were poured on the 250 μm sieve. The block was horizontally agitated for

approximately 2 minutes at 125 rpm while deionized water was poured until Magg disaggregated completely. The material retained in the 250 μm and 50 μm sieves corresponded to the fractions of > 250 μm (cPOM + sand), and mMagg (50-250 μm), respectively. Similar to the first step, the M(s+c) fraction was recovered by centrifugation. The three fractions were dried at 50 ºC and stored at ambient temperature.

## 2.5 Other soil indicators

As a complement of the detailed study of water retention, soil porosity and structure, other indicators related to soil functioning and agricultural sustainability were analysed. First, the distribution of organic C among aggregate fractions was determined by analysing the organic C concentration in every fraction by wet oxidation following (Tiessen and Moir, 1993). Second, microbial biomass C (MBC) was measured by fumigation-extraction as described by (Vance et al., 1987), and the functional diversity of the soil microbial populations was carried out following (Preston-Mafham et al., 2002) from fresh samples, and

by a study of the utilization patterns of different C sources with Ecoplates[TM] (Biolog, Hayward, CA, USA). The average well color development (AWCD) and the number of substrates used by the microbial community within the soil (NSU) were

determined from the Ecoplates™, as quantitative indicators of the soil functional diversity based on community-level physiological profiles (Zak et al., 1994).

## 2.6 Statistical analysis

Three (n=3) replicates of each study treatment (OPM and CM) were used in the statistical analysis. A one-factor analysis of variance (ANOVA) with significance level $p < 0.05$ was performed for the different indicators to examine the significant influence of OPM. All statistical analysis were performed using IBM SPSS Statistics 27.0 (SPSS Inc., 2021).

## 3. Results

### 3.1 Analysis of the SWRCs

A clear difference between the SWRCs of the two treatments was observed: the variability between treatments was remarkably superior to the one existing between the replicates of the same treatment (Fig. 1).

**Figure 1. Soil water retention curves for each replicate (n= 3) for the two treatments (optimized management (OPM) vs. Conventional management (CM).**

The saturation water content in both treatments was similar ($p > 0.05$), which indicates that there was no significant compaction
(and therefore, reduction of the total porous space) because of management for the studied depth. This is consistent with the observation of soil in both treatments presenting the same bulk density (Table 1).

In relation to the shape of the SWRCs, both corresponded to the S type (Kosugi et al., 2002): a relevant presence of occluded or non-functional pores was not observed (the air-entry region was negligible, Fig. 1).

Nonetheless, the specific water capacity –change in the moisture content per unit of suction; $d\theta/d\Psi$, as defined by (Klute,
1952)– in the suction range between saturation (0 kPa) and near field capacity ($10.5 \pm 0.56$ kPa) was significantly higher for CM ($d\theta/d\Psi= 1.89 \pm 0.32$) than for OPM ($d\theta/d\Psi= 0.34 \pm 0.05$). However, when suction was greater than 10 kPa, the value of specific water capacity tended to be similar for both treatments, with no significant differences ($p > 0.05$) above 32 kPa ($d\theta/d\Psi= 0.10 \pm 0.01$) (Fig. 1).

### 3.2 S index

The S-index obtained from both the van Genuchten equation (Table 2) and the *ad hoc* sigmoidal equation (Table 3) showed no significant differences ($p > 0.05$) between both treatments. However, it should be noted that the S values obtained from the van Genuchten equation showed a better performance, with a dispersion one order of magnitude smaller than that obtained from the *ad hoc* sigmoidal equation.

The S value for the two study treatments reflected *good* soil physical quality ($0.035 < S \leq 0.050$) for the van Genuchten equation (Table 2) and *very good* ($\geq 0.050$) for the sigmoidal equation (Table 3) (Bacher et al., 2019; Dexter, 2004b; Reynolds et al., 2009).

**Table 2. S index values, contents of water ($\theta$) and suction ($\Psi$) corresponding to the inflection point, obtained with the van Genuchten equation, and van Genuchten parameters. Mean ± standard deviation of the mean (n=3). All the differences are not statistically significant (p > 0.05).**

**Table 3. S index values and contents of water ($\theta$) and suction ($\Psi$) corresponding to the inflection point, obtained with the sigmoidal equation adjusted to experimental data. Mean ± standard deviation of the mean (n=3). Statistically significant differences (p < 0.05) are in bold.**

### 3.3 Bi-modal approach

Experimental results were plotted as differential functions [$d\Theta/d(\log h)$ vs $\log h(h)$] seeking for a multimodal behavior: all the curves analyzed seemed to be of the uni-modal type (data not shown).However, it should be noted that suction values did not exceed 150 kPa, and according to Dexter et al. (2008) (cf. their Fig. 3) and Jensen et al. (2019) (cf. their Fig. 2) findings the second peak defining a bimodal behavior seems to appear at suction around 1000 kPa. Then, we tried again incorporating to the dataset the water content-suction measurements at 1500 kPa with the same result, i.e. unimodal behavior. But this could be an artifact of the dataset since there is a wide experimental gap between 150 kPa and 1500 kPa, i.e. no measurements in between.

Despite this, the double-exponential equation for soil water retention proposed by Dexter et al. (2008) was explored (Eq. 3) (Table 4). The structural pore space would have been reduced by 35% as a result of no-tillage (OPM) (cf. $A_2$ values, Table 4), while the matrix pore space values remain rather constant in both treatments (cf. $A_1$ values in Table 4).

 **Table 4. Average values of the fitted parameters of the double-exponential water retention equation by Dexter et al. (2008) obtained with the experimental dataset. Mean ± standard deviation of the mean (n=3). All the differences are not statistically significant (p > 0.05).**

### 3.4 WRa index

The soil under OPM (WRa= 4.6 ± 0.5; average ± standard deviation) seemed to have a better structure than the soils under
CM (WRa= 4.1 ± 1.1) because the former held the same relative fraction of water with more absolute energy in its porous system (Armindo and Wendroth, 2016). However, this difference between treatments was not statistically significant due to the large variability observed in the CM treatment.

### 3.5 Analysis of the pores size-distribution

Fig. 2 depicts the probability distribution function of pore size (mean of the three replicates) for the study soil under OPM and
CM, and the classification of pore sizes according to the Soil Science Society of America (Weil and Brady, 2017).

**Figure 2. Probability distribution function of pore size (mean of three replicates) of the soil under the two studied treatments (OPM and CM), and pore size classification (Weil and Brady, 2017). Note: X-axis in logarithmic scale.**

For both treatments, the percentage of mesopores (equivalent diameter between 30 and 80 μm) was similar (5.6 ± 0.7 in OPM and 8.0 ± 1.3 in CM) ($p > 0.05$). Similarly, the population of smaller pores (micropores, with equivalent diameter between 5
and 30 μm) did not present significant differences for both treatments (15.5 ± 1.1% in OPM and 16.4 ± 2.3% in CM, Fig. 2) ($p > 0.05$), which confirmed the textural homogeneity of the soil in both treatments (Table 1), as this porosity is more associated with soil texture than the soil structure (Pagliai et al., 2004).

On the contrary, the proportion of pores with equivalent diameters > 80 μm (macropores) differed between treatments ($p <$
0.05). For CM, macropores represented 27.7 ± 4.8% of total porosity and only 11.6 ± 2.3% for OPM. As such, in CM, the
population of pores with equivalent diameter 500-1000 μm and > 1000 μm represented 5.5 ± 1.3% and 4.4 ± 2.2%, respectively.
For OPM, the population of pores larger than 500 μm –considered mainly as fissures (Pagliai et al., 2004)– was 2.8 ± 1.3%, with no apparent presence of pores larger than 1000 μm (< 1.3 ± 0.7%). As macropores drain quickly at low suctions, when these macropores become empty, the volume of soil capable of storing available water is then reduced. Therefore, up to 100 kPa, the soil under OPM could eventually store a higher amount of water (ca. 10-15%) per unit of volume than under CM (Fig.
1).

### 3.6 Analysis of the size-distribution of stable aggregates

Mass losses during fractionation accounted for $3.6 \pm 0.2\%$ of the initial samples, with no differences between treatments (data not shown), which means that the differences found (Fig. 3) can be considered as a response to the studied treatments.

For both treatments, the percentage of soil within water-stable macro (Magg) or micro (magg) was $92.2 \pm 0.3\%$, and the non-aggregated (s+c) fraction presented $5.8 \pm 0.4\%$ of the initial mass (Fig. 3). Within the aggregated fractions (Magg + magg), clear differences were observed in the size-distribution of aggregates ($p < 0.05$): the soil under OPM had $75.9 \pm 2.6\%$ of stable macroaggregates (Magg, > 250 µm); while this percentage was of $57.5 \pm 2.1\%$ for CM.

In relation to the composition of Magg, both cPOM and mMagg represented a greater proportion of Magg in OPM in comparison to CM (where M(s+c) represented a greater proportion of total Magg mass) (Fig. 3). It has to be noted that both cPOM and mMagg included an undetermined percentage of sand particles. However, the similar texture of the soil for both treatments (Table 1) allows to consider that the observed differences cannot be attributed to differences in the sand content.

**Figure 3. Size-distribution of stable aggregates and individual particles in the soil under OPM (a) and CM (b) Magg: Macroaggregates; magg: microaggregates; mMagg: microaggragates within macroaggregates. s+c: silt+clay fraction; cPOM: coarse particulate organic matter >250 µm and sand particles. The error bars represent the standard error, which is the standard deviation divided by the square root of the sample size. All aggregate fractions are significantly different (p < 0.05) between OPM and CM, with the exception of mMagg.**

### 3.7 Organic C storage and soil microbial diversity

The distribution of soil organic C (SOC) among aggregate fractions is shown in Fig. 4. It is worth mentioning that, for the two management systems, the carbon recovery data after fractionation were satisfactory, since no more than 10% of the initial soil C was lost during the fractionation procedure (data not shown).

**Figure 4. Distribution of organic C in stable aggregates and individual particles in soil under OPM (a) and CM (b) Magg: Macroaggregates; magg: microaggregates; mMagg: microaggragates within macroaggregates. s+c: silt+clay fraction; cPOM: coarse particulate organic matter >250 µm and sand particles. The error bars represent the standard error, which is the standard deviation divided by the square root of the sample size. All aggregate fractions are significantly different (p < 0.05) between OPM and CM, with the exception of s+c and mMagg.**

In our study, soil management resulted not only in higher SOC concentration under OPM (Table 1), but also in a different distribution of SOC among aggregate size fractions. As such, OPM resulted in a higher proportion of SOC stored in Magg ($77.7 \pm 2.9$ g C 100 g$^{-1}$ soil C) than CM ($61.1 \pm 2.2$ g C 100 g$^{-1}$ soil C). Conversely, CM contained proportionally more SOC

in magg and s+c, < 50 μm fractions. The greater proportion of SOC accumulated in Magg corresponded to that found in cPOM > 250 μm (30.2 ± 2.2 g C 100 g$^{-1}$ Magg-C in OPM for 11.1 ± 1.4 g C 100 g$^{-1}$ Magg-C in CM).

In relation to the soil microbiological indicators, OPM did not only result in more MBC, but also in a higher efficiency for the degradation of organic substrates (degrading 29.17% more substrates than in the conventional system (NSU, Table 5)). Likewise, a more intense degradation of the substrates (> AWCD) was observed under OPM than CM.

**Table 5. Biological indicators. Mean ± standard deviation of the mean (n=3). Statistically significant differences (p < 0.05) are in bold.**

Finally, since this work was conducted in farmers' plots, yields were not explicitly measured as it is usually done in experimental fields, but some basic data are available from the farmers managing the fields (see Table 6 below). From these data, no apparent differences between treatments in crop yields occurred in the study area.

**Table 6. Average crop yield (2016-2021) of OPM and conventional agricultural fields under conventional tillage (CM), as reported by farmers.**

## 4. Discussion

Dexter and Bird (2001) stated that one of the applications of the S-index was to identify the optimal water content for tillage, which would correspond to the inflection point of the SWRC. This is in agreement with our results: moisture contents corresponding to S values were all near field capacity (Tables 2 and 3), water content at which tillage produces the greatest proportion of small clods, which can be considered an achieved tillage.

Despite this observation, the S-index was not sensitive enough to reflect differences in the soil physical quality due to the different soil and crop managements assessed. This, despite the high degree of detail of the SWRCs used, which facilitates an optimal adjustment of the different mathematical functions applied. Alonso et al. (2022) also found no significant differences in S-index values between silt loam and sandy loam soils subjected to mouldboard plowing, deep loosening and minimum tillage managements, while other soil physical quality variables did show significant differences between those soils. The S-index is probably aimed at comparing soils in more contrasting conditions, especially in terms of bulk density, texture and organic matter content, as inferred from the case studies presented by Dexter, (2004a).

The differences in soil quality between the two forms of management (OPM vs CM) were better observed from the pore size distribution –obtained from the SWRCs– and by the analysis of size-distribution and stability of soil aggregates. To this respect, our results showed an overabundance of macropores (> 80 μm) under CM while the amount of mesopores (available water) and micropores were similar in both treatments (Fig. 2). In other works in which SWRCs were used for the long-term study of

pore size distribution in no-tillage (NT) and conventional tillage (CT) management, it was found that there is no unanimity in the results obtained (Wardak et al., 2022). Pires et al. (2017) evaluated the effect of tillage and direct seeding on the structure of an Oxisol through the analysis of SWRCs and micromorphological assessments. From their results, it can be observed (see Fig. 1 and 2, Pires et al., 2017) that the soil under conventional tillage reduced its water content (starting from saturation) by 15% when a suction of approximately 20 cm was applied. For the soil under direct seeding, the decrease was only of 5%. For the depth range 10-30 cm, the changes in moisture content with suction were similar for both treatments. In addition, in the soil under direct seeding pores within the size range 50-500 μm –responsible for draining excess water (Greenland and Pereira, 1977)– occupied 39% of the total porous space, while for tilled soil the percentage was slightly over 60%. Lipiec et al. (2006) observed that the pore system of a silty clay loam soil under CT presented greater macroporosity, with the differences between tillage treatments being more pronounced in the 0-10 cm depth than in the 10-20 cm depth. Similar results were obtained in clayey soils by Tuzzin de Moraes et al. (2016) and Borges et al. (2019), who identified significantly higher macroporosity in CT treatment compared to NT. In contrast, Imhoff et al. (2010) and Gao et al. (2019) observed increased macroporosity in the NT treatment in a silty loam soil and a sandy loam soil, respectively.

In addition, in the study of micro- and mesopores, most studies have observed an increase of these pores under NT compared to CT. Examples are the works of Borges et al., (2019), Lipiec et al. (2006) and Tuzzin de Moraes et al. (2016), whose analysis of SWRCs showed a higher volume of micropores and mesopores under NT than under CT. However, in the study by Imhoff et al. (2010) a decrease in micro- and mesopore volume under NT was recorded. Similarly, Gao et al. (2019) saw reduced mesoporosity in NT soil, observing no significant effect of such reduction on the soil hydraulic properties. It has to be noted that these pores are relevant for soil functioning, as mesopores are associated with water retention after free drainage, with a suction that enables easy extraction by plants (available water), transmitting water by capillarity to the radicular zone (Weil and Brady, 2017).

In any event, it seems from this variability of results, that the impact on soil management on soil porosity is site-dependent. In agronomic and climatic conditions closer to the soil studied here, Pagliai et al. (1984) studied the size distribution and shape of pores in a clay loamy vertic soil under CT and NT, using micromorphological image analysis of soil thin sections. The size-distribution of pores was more regular in the soil under direct seeding than under conventional tillage. For direct seeding, 7% of the total pores identified (=145) were macropores (500-1000 μm), occupying 25% of the porous space (image area). For conventional tillage, in turn, the bias was considerable: 22% of total pores (=45) corresponded to macropores (500-1000 μm and >1000 μm), occupying approximately 85% of the porous space (Fig. 1, Pagliai et al., 1984). This greater macroporosity in OPM can explain the fast desorption rate at low suction values (high specific water capacity) observed in CM compared with OPM (Fig. 1).

The overabundance of macropores in soils under CM in our study could be to some extent explained by an increase in soil fragments rather than soil aggregates in the CM in comparison with OPM treatment. Soil aggregates and fragments may look

similar but are formed by different processes and have different properties (Or et al., 2021): soil fragments form by mechanical forces of tillage; they tend to be mechanically weak and coalesce upon wetting with macroposity collapsing within a single season. Instead, soil aggregation is stimulated by biological activity with biopolymers and hyphae that stabilize and bind soil particles. In short, soil aggregates are more stable than soil fragments. Borges et al. (2019) observed significantly higher macroporosity in a soil under conventional tillage compared to a soil under minimum tillage; they explained this to the mechanical action of tillage. The non-bimodal behavior of our SWRCs did not allow to verify this extent from the $A_2$ values (Dexter et al., 2008), theoretically corresponding to the structural pore space (Table 4).

In relation to aggregation, Fuentes-Guevara et al. (2022) found a significant correlation between hydraulic-energy based indices –including WRa– with some physical properties before and after land leveling operations, indicating their capacity to capture soil structure changes. The high variability observed for this index in CM (section 3.4) hindered however their use for such an assessment in our case. However, the preponderance of Magg under OPM (Fig. 3) can be understood as a consequence better soil condition (or lower degradation) than under CM, in terms of aggregates stability. As conceptualized in the hierarchical model of soil aggregation (Angers et al., 1997; Beare et al., 1994; Golchin et al., 1994; Oades, 1984; Six et al., 1999, 2004; Tisdall and Oades, 1982), while magg are formed within Magg, and stabilized mostly by the action of persistent agents (e.g., cationic complexes, humidified organic matter), Magg are stabilized by the action of transitory agglutinating agents (hyphae and mycorrhizae, microbial and vegetable derivatives). The main implication of this hierarchy is that agricultural management primarily affects the less stable macroaggregates, while the more stable microaggregates are less influence. Implicit in this concept is the fact that aggregates form sequentially (Jarvis, 2012). According to this hierarchical vision of soil aggregation, these agglutinating agents are, in turn, widely conditioned by soil management: the formation of (macro) aggregates is thus favoured by the lower degree of soil disturbance by tillage, higher inputs of crop (organic) residues in the soil organic matter pool, and the punctual organic amendments used in OPM (Jastrow, 1996; Lehmann and Kleber, 2015; Six et al., 2004; Tisdall and Oades, 1982). This observation is supported by the higher proportion of cPOM (Fig. 3) and total organic C (Table 1) found under OPM, as explained below.

Although the relationship between organic matter cycling and soil structural stabilization has been observed to be soil-dependent (Rasmussen et al., 2018), and the calcareous nature of the studied soil may interact with it by stabilizing Magg and magg to a greater extent than in Ca-free soils (Fernández-Ugalde et al., 2011; Rowley et al., 2018, 2021), the greater accumulation of SOC within stable Magg in OPM than CM (Fig. 4), suggests that the response of soil structure to the reduction of tillage and the increase in organic C inputs corresponded to that observed previously in other soil types (Six et al., 2004; Fernández-Ugalde et al., 2016), and in soils of the same type in the region (Virto et al., 2007; Yagüe et al., 2016).

In relation to our objectives, these results indicate that the changes observed in the physical soil indicators studied above can be related to a more positive SOC balance, very likely related to more inputs from vegetation cover and fewer interruptions of the SOC cycling due to tillage. Soil C storage is generally observed as a key soil property, related to both soil functioning and

the global C cycle. As such, it has been proposed as an indicator for several soil functions, including nutrient recycling, functioning of soil ecosystems, pollution control, food security and global change (Paul, 2016).

In addition, the accumulation of cPOM, which has been repeatedly identified as a fast cycling pool, and a precocious indicator of changes in SOC cycling (Cotrufo et al., 2019), can be understood as the result of SOC cycling being more active, and resulting in a greater proportional accumulation of labile forms of SOC under OPM than CM (Lehmann and Kleber, 2015).

The idea of a more active SOC cycle under OPM was supported by the observed higher microbial activity and microbial biomass C under OPM compared to CM (Table 5), which can be associated with better conditions for SOC degradation and stabilization under OPM (Six et al., 2002).

Other relevant consequences of the observed results in the topsoil of the studied sites can be those related to the control of soil losses through erosion. This depends, among other factors such as ground cover (granted by OPM), on the soil own resistance to slaking and aggregates breakdown, and on the infiltration rates. Greater resistance of aggregates was clearly observed in OPM in our study, suggesting reduced erodibility. This supports the view of the use of cover crops in sensitive areas (Panagos et al., 2021) as a useful tool for the involvement of farmers in the reduction of erosion rates (Panagos et al., 2021; Mosavi et al., 2020; Grillakis et al., 2020; Eekhout and De Vente, 2020; Paroissien et al., 2015). In addition, although the assessment of water infiltration and hydraulic conductivity of the soil in field conditions are beyond the scope of this work, and without other consideration such as the possible existence of compacted layers at depth caused by tillage (Fernández-Ugalde et al., 2009), this suggests a faster infiltration of water under this treatment. Considering the vulnerable character of this area with respect of groundwater pollution by nitrates, this would indicate a worse condition of soils in the area under conventional practices in terms of reaching the environmental goals in relation to fresh water quality set by the EU (Fetting, 2020), the UN (Rattan et al., 2018) and other national and regional environmental policies.

Finally, it has to be noted that agricultural sustainability cannot forget the interest of farmers. Although yield data were available only from indirect sources (Table 6), they suggest that the implementation of OPM did not imply a relevant reduction of yields in the study area, as is often observed when reduced input strategies are introduced in some agrosystems. For instance, Or et al. (2021) have reported an average reduction of 10% yields upon NT adoption.

In summary, and from a general point of view of the sustainability of agricultural management and the multifunctionality of soils (Bouma et al., 2019), these results indicate that OPM did not only result in differences in water retention and soil structure that can contribute to improve water-use efficiency and crops productivity, but also in enhanced biodiversity and increased SOC storage. OPM seems from this perspective, a useful tool in face to the present challenges and commitments of agriculture in Europe and worldwide (Bouma et al., 2022; Panagos et al., 2022).

## 5. Conclusions

A pioneer and unique in the region (Navarre, Spain) optimized soil and crop management system that includes, among other techniques, reduced tillage, crop rotations, and the occasional application of organic amendments was assessed for the soil physical quality after 18 years of its implementation. Our findings suggest, first, that some classical approaches to the assessment of SWRCs cannot capture the actual consequences of the use of these optimized management strategies on soil quality. However, detailed SWRCs were seen useful to identify relevant changes in soil porosity.

In relation to the physical quality of the soil, the innovative management tested here provided good results after 18 years – highlighting the proportion and size of water-stable soil macroaggregates–. It also contributed to a more abundant and diverse soil microbial population, and could contribute to an improvement in water use efficiency. This is especially relevant for rainfed agriculture where water is the most limiting factor for crops growth, such as the study zone.

The optimized management analyzed herein can therefore be recommended for higher soil sustainability in Mediterranean agrosystems. However, it is not currently a widespread practice in the region; most likely because the high initial investment and the farmer's concern that crop yields would be reduced. This work illustrates the need for an adequate assessment and dissemination to overcome these reluctances of farmers and other potential barriers to the adoption of this type of systems.

Further analysis at deeper soil layers –at least to the rooting depth– are necessary for a more complete assessment of the proposed optimized management. Moreover, to better understand changes in the soil hydrology, it is necessary to carry out experiments to determine infiltration rates, preferably under controlled suction. Finally, future studies should take a dynamic approach to soil water regimes by taking advantage of the widely available dynamic simulation models of the soil-water-atmosphere-plant system.

**Data availability.** The data that support the findings of this study are available from the corresponding author upon reasonable request.

**Author contribution.** RG and IV conceptualized and supervised the paper. LA designed the experiments and AA carried them out. RG, MC and AA visualized the project, did the formal analysis, and conducted the investigation with IV, who also collected the resources. AA prepared the manuscript with contributions from all co-authors.

**Competing interests.** The contact author has declared that neither herself nor her co-authors have any competing interest.

**Financial support.** This study was developed within the framework of project 011-1365-2020-000075 CropStick: sentinel of salts, pH, nitrogen and nutrients, and deep percolation, financed by the Government of Navarre.

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

**Table 5. Physical-chemical properties of the soil (0-30 cm) in OPM and CM treatments and the textural characterization of both treatments. Mean ± standard deviation of the mean (n=3). Statistically significant differences (p < 0.05) are in bold.**

| Treatment | Optimized (OPM) | Conventional (CM) |
|---|---|---|
| Bulk density (0-5 cm) (g·cm$^{-3}$) | 1.26 ± 0.05 | 1.26 ± 0.15 |
| pH | 8.00 ± 0.05 | 8.01 ± 0.01 |
| Organic C (%) | **1.80 ± 0.10** | **1.51 ± 0.14** |
| CE (µS·cm$^{-1}$) | **483 ± 5.66** | **795 ± 4.24** |
| Carbonates (%) | 31.6 ± 0.19 | 32.5 ± 0.14 |
| Sand (Coarse) (%) | 5.05 ± 0.08 | 5.79 ± 0.33 |
| Sand (Fine) (%) | 30.9 ± 1.00 | 31.7 ± 1.25 |
| Silt (%) | 47.2 ± 1.23 | 43.7 ± 0.93 |
| Clay (%) | **16.9 ± 0.46** | **18.5 ± 0.46** |
| **Texture class (USDA)** | *Loam* | *Loam* |


**Table 6. S index values, contents of water (θ) and suction (Ψ) corresponding to the inflection point, obtained with the van Genuchten equation, and van Genuchten parameters. Mean ± standard deviation of the mean (n=3). All the differences are not statistically significant (p > 0.05).**

| | S index | Inflection point | | van Genuchten parameters | | | |
|---|---|---|---|---|---|---|---|
| | | $\theta_g$ (%) | $\Psi$ (kPa) | $\alpha$ | n | m | $\theta_{sat}$ (%) |
| **OPM** | 0.035 ± 0.002 | 33.80 ± 4.71 | 23.85 ± 20.19 | 0.07 ± 0.07 | 1.11 ± 0.02 | 0.10 ± 0.02 | 42.96 ± 0.05 |
| **CM** | 0.035 ± 0.007 | 31.66 ± 3.80 | 6.22 ± 4.01 | 0.18 ± 0.15 | 1.12 ± 0.03 | 0.11 ± 0.02 | 40.73 ± 0.05 |


**Table 7.** S index values and contents of water (θ) and suction (Ψ) corresponding to the inflection point, obtained with the sigmoidal equation adjusted to experimental data. Mean ± standard deviation of the mean (n=3). Statistically significant differences (p < 0.05) are in bold.

| Treatment | S index | Inflection point | |
|---|---|---|---|
| | | $\theta_g$ (%) | Ψ (kPa) |
| **Optimized (OPM)** | 0.040 ± 0.016 | **36.53 ± 0.98** | 6.97 ± 6.70 |
| **Conventional (CM)** | 0.057 ± 0.012 | **29.57 ± 0.81** | 9.53 ± 2.15 |


**Table 8. Average values of the fitted parameters of the double-exponential water retention equation by Dexter et al. (2008) obtained with the experimental dataset. Mean ± standard deviation of the mean (n=3). All the differences are not statistically significant (p > 0.05).**

| Treatment | Parameters of the Dex model | | | | | |
|---|---|---|---|---|---|---|
| | C | $A_1$ | $h_1$ | $A_2$ | $H_2$ | RMSE |
| | $m^3 \cdot m^{-3}$ | $m^3 \cdot m^{-3}$ | hPa | $m^3 \cdot m^{-3}$ | hPa | $m^3 \cdot m^{-3}$ |
| **OPM** | $0.25 \pm 0.04$ | $0.11 \pm 0.02$ | $865 \pm 495$ | $0.06 \pm 0.04$ | $29.9 \pm 21.9$ | $0.005 \pm 0.005$ |
| **CM** | $0.20 \pm 0.01$ | $0.11 \pm 0.02$ | $737 \pm 332$ | $0.10 \pm 0.03$ | $25.2 \pm 9.83$ | $0.003 \pm 0.001$ |


**Table 5. Biological indicators. Mean ± standard deviation of the mean (n=3). Statistically significant differences (p < 0.05) are in bold.**

| Biological indices | Optimized (OPM) | Conventional (CM) |
|---|---|---|
| MBC (mg C/kg soil) | **518 ± 35.2** | **318 ± 35.2** |
| NSU | **22.0 ± 0.58** | **17.0 ± 2.08** |
| AWCD | **0.79 ± 0.03** | **0.64 ± 0.06** |

**Table 6. Average crop yield (2016-2021) of OPM and conventional agricultural fields under conventional tillage (CM), as reported by farmers.**

| | Yields (t/ha) | |
| --- | --- | --- |
| **Crop** | **OPM** | **CM** |
| **Wheat** | 6.8 - 9.3 | 5.5 - 7.0 |
| **Barley** | 5.8 - 8.0 | 5.0 - 6.5 |
| **Rapessed** | 2.0 - 4.0 | 2.0 - 3.0 |
| **Legumes** | 2.2 - 3.5 | 1.7 - 2.5 |

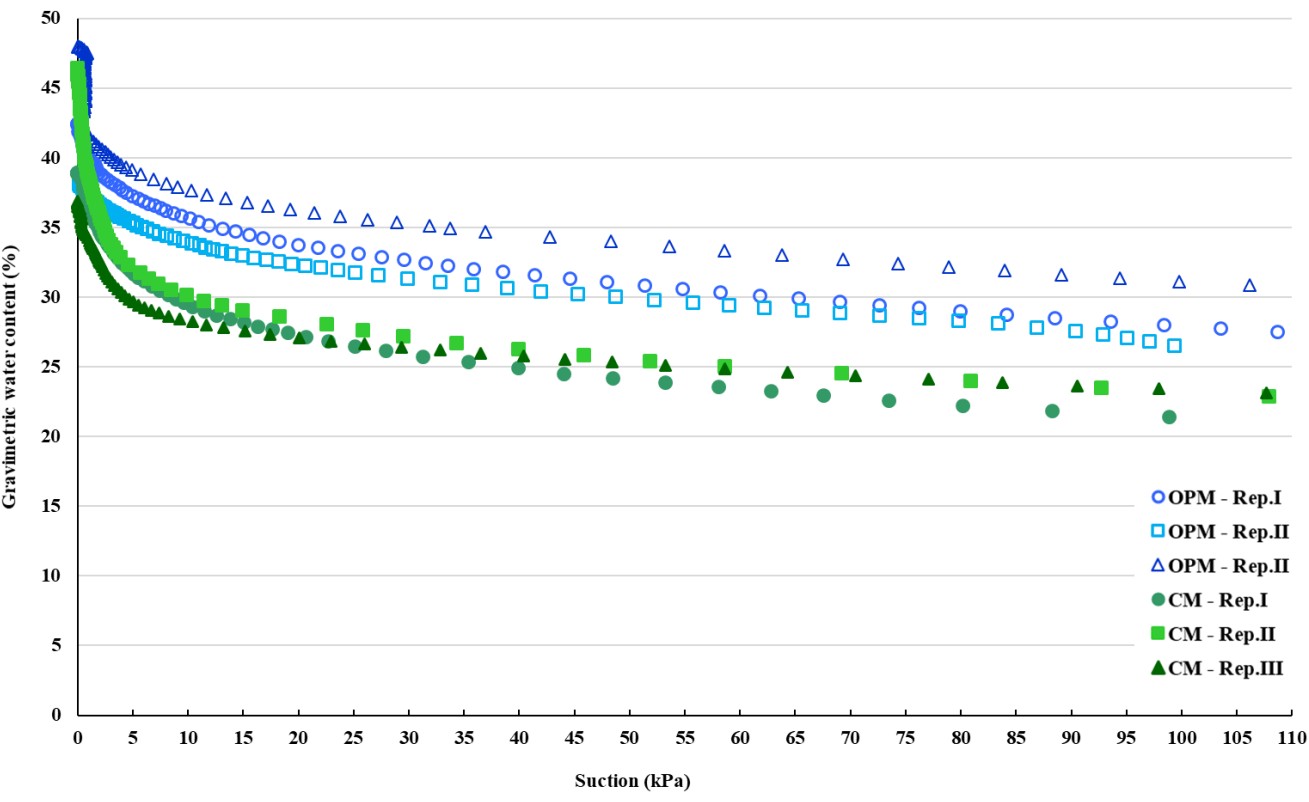


**Figure 1. Soil water retention curves for each replicate (n= 3) for the two treatments (optimized management (OPM) vs. Conventional management (CM).**

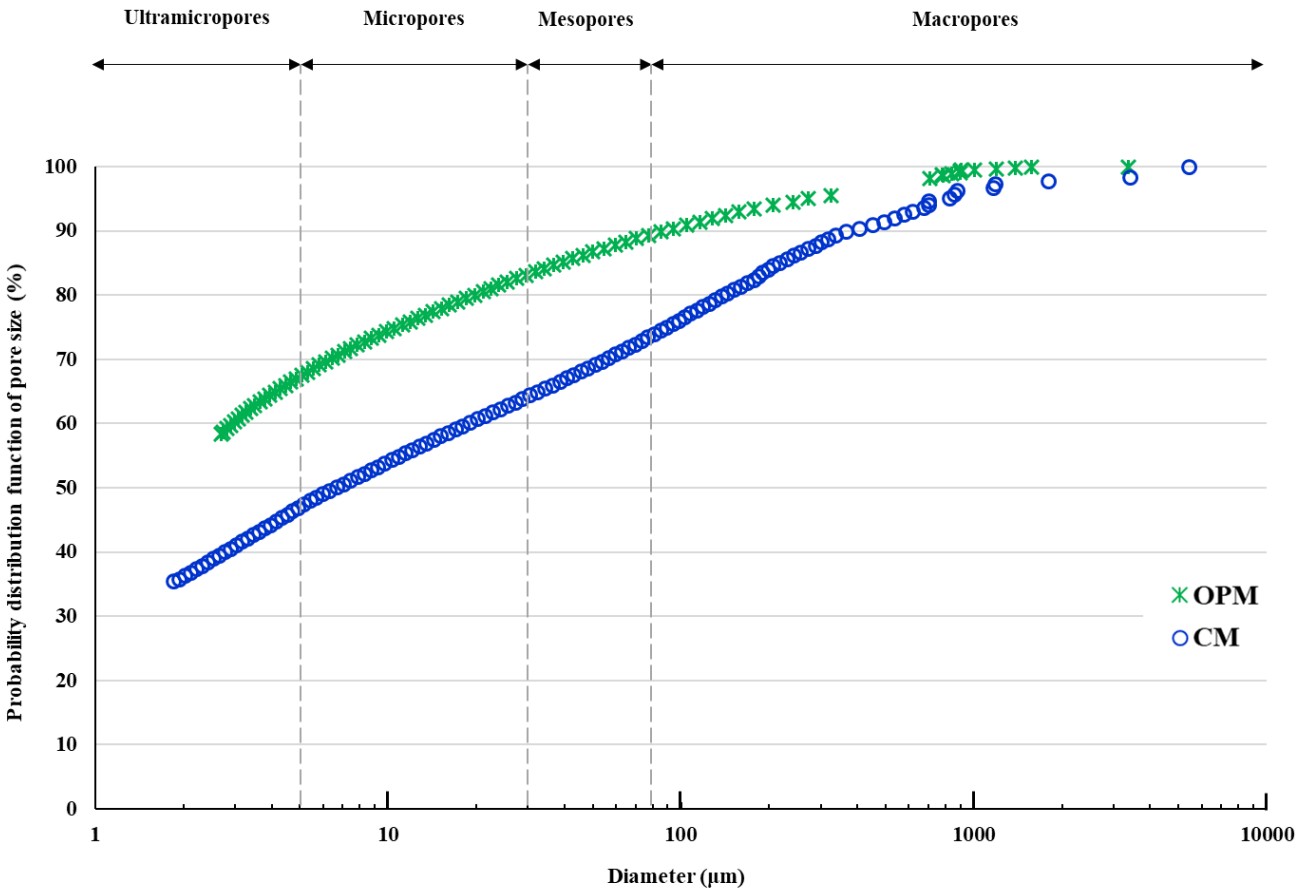


**Figure 2. Probability distribution function of pore size (mean of three replicates) of the soil under the two studied treatments (OPM and CM), and pore size classification (Weil and Brady, 2017). Note: X-axis in logarithmic scale.**

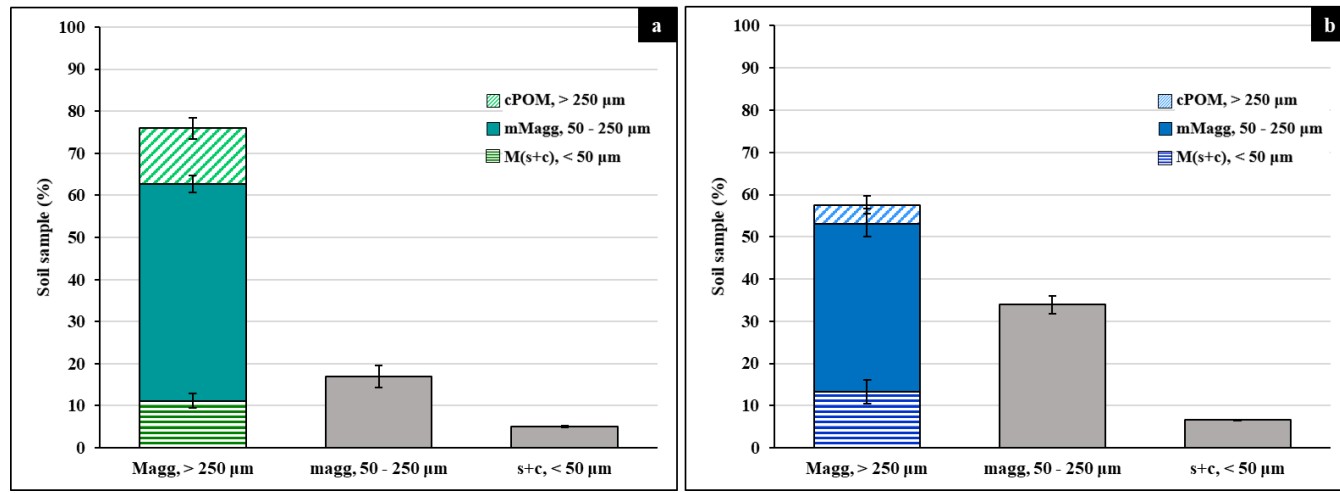


Figure 3. Size-distribution of stable aggregates and individual particles in the soil under OPM (a) and CM (b) Magg: Macroaggregates; magg: microaggregates; mMagg: microaggragates within macroaggregates. s+c: silt+clay fraction; cPOM: coarse particulate organic matter >250 µm and sand particles. The error bars represent the standard error, which is the standard deviation

divided by the square root of the sample size. All aggregate fractions are significantly different (p < 0.05) between OPM and CM, with the exception of mMagg.

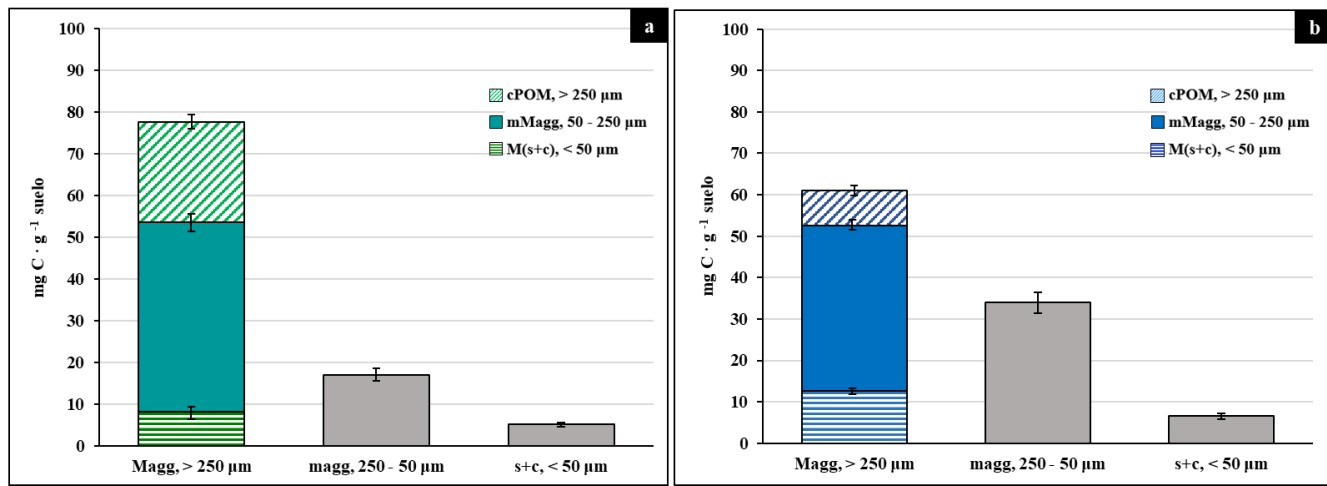

**Figure 4. Distribution of organic C in stable aggregates and individual particles in soil under OPM (a) and CM (b) Magg: Macroaggregates; magg: microaggregates; mMagg: microaggragates within macroaggregates. s+c: silt+clay fraction; cPOM: coarse particulate organic matter >250 µm and sand particles. The error bars represent the standard error, which is the standard deviation divided by the square root of the sample size. All aggregate fractions are significantly different (p < 0.05) between OPM and CM, with the exception of s+c and mMagg.**
