# Peer review of "Effects of innovative long-term soil and crop management on topsoil properties of a Mediterranean soil based on detailed water retention curves"

_EGUsphere, 2022_

## Author Comment (AC1)

This reviewer strongly supports the type of research that the authors present in their paper: characterize field conditions on soils where certain management types have been applied for a substantial period. The authors also present a professional paper in terms of methods used, particularly the method to measure retention curves and by applying multiple experiments and proper statistical analyses.

Unfortunately, when they conclude in the end that: "*the study contributes to higher sustainability of mediterranean agrosystems*", they are off the mark. Sustainable development is defined as having economic, social and environmental dimensions as expressed by the UN Sustainable development Goals (SDG) and the associated European Green Deal. Scientific papers published in 2022 that suggest a link with sustainable development cannot ignore these developments in the scientific, policy and public arena's. The title of the paper is also highly misleading: "*improvement of cultivated soil*" cannot be based only on: "*water retention curves*". Numerous published papers describe a systems analysis based on the interaction between soils-water-atmosphere-plants that is needed to assess effects of soil management.

Sustainable development implies for agriculture at least: production of healthy food ( SDG2&3), protection of ground- and surfacewater quality (SDG6), carbon capture and reduction of greenhouse-gas emission (SDG13) and increasing biodiversity and combatting land degradation (SDG 15).

Firstly, thank you very much for your thorough review and your helpful comments and suggestions that significantly helped to improve our manuscript.

We agree with the reviewer that our findings are too narrow to be a comprehensive contribution to the sustainability of Mediterranean agrosystems considering all the aspect that sustainability development really involved (SDG), as clearly remarks the reviewer. In fact, our main objective, as well pointed out by the reviewer, is the evaluation of an innovative -and pioneer in our region (Navarre)- soil and crops management strategy on topsoil properties. Indeed, to our knowledge, there is no other agricultural field in the whole region of Navarre where soil and crop management as proposed herein (OPM) is practiced and even less for almost two decades, with the exception of our small OPM test area were a pioneer farmer works for almost 20 years. This represents in our view, the uniqueness of our OPM study. This relevance of our work would be better and clearly pointed out in the text, especially in the Introduction and in section 2.1. (Study site [former 'zone'] and treatments).

Our evaluation is carried out through the analysis of soil water retention (curves) and also soil structural stability, as they can be considered indicators of the two most relevant limitations for sustainable crop production in the area, namely water deficit (usually expressed as annual drought) and soil loss. In short, we characterize two "phenoforms" of a typical Mediterranean soil ("genoform"). Ultimately, this is just a first step towards a more complete evaluation of the proposed agrosystem.

We believe that the actual scope of our work could be unambiguously presented as follows. First, deleting the questioned statement that "the study contributes to higher sustainability of Mediterranean agrosystems". Second, reformulating the title following the reviewer suggestion: *Effects of innovative long-term soil management on topsoil properties of a Mediterranean soil based on detailed water retention curves.*

Moreover, our work can be improved by assessing other indicators related to the SDG as indicated below.

**I would recommend that the authors frame their results in a SDG context:**

1. It should not be too difficult to report crop yields. (SDG2&3).

As this work was conducted in farmers' plots, yields were not explicitly measured as it is usually done in experimental fields, but some basic data are available from the farmers managing the fields (see Table 1 below). From the data provided by the farmers we cannot infer assure that the OPM treatment was better than the CM in terms of yield. Generally, there is usually no difference between the two treatments because the limiting factor is mostly water. Even so, the data suggest that farmers are not going to see a decrease in the yields of their plots by implementing sustainable practices, i.e. OPM. This is an interesting finding since normally no-till often results in reduction in crop yields of ca. 10 % (Or et al., 2021).

Table 1. Average crop yield (2016-2021) of OPM and conventional agricultural fields under conventional tillage (CM), as reported by farmers.

| Crop | Yields (t/ha) | |
| --- | --- | --- |
| | OPM | CM |
| Wheat | 6.8 - 9.3 | 5.5 - 7.0 |
| Barley | 5.8 - 8.0 | 5.0 - 6.5 |
| Rapessed | 2.0 - 4.0 | 2.0 - 3.0 |
| Legumes | 2.2 - 3.5 | 1.7 - 2.5 |

2. Water tables may be deep and not polluted; this could be mentioned showing that groundwater quality is not really an issue here.(SDG6).

We appreciate this observation in particular, because groundwater quality is in fact an issue as the area is within a zone classified as vulnerable to nitrate contamination of the watertable as defined by the EU Nitrates Directive (Council Directive 91/676/EEC) (Council of the European Union, 2008). With this, and considering our results of a lower proportion of draining macropores under the optimized management strategy considered, we could infer that its adoption can also be an asset in terms of groundwater protection. However, we cannot be conclusive on this issue without a better understanding of the soil water dynamics.

3. Carbon capture is not evident as the % C is about the same for OPM and CM. This is interesting because OPM management is supposed to lead to higher % C and their results indicate this may apply to soils in humid regions ,as reported in literature, but not in arid climates. Effect of high temperatures? But.. data only are restricted to 30 cm depth. (SDG13).

The reviewer is right and generally, OPM management is usually associated with an increase in C storage compared to CM. Nevertheless, it has been seen that when tillage is suppressed and crop yields are maintained under OPM and CM, as in our study, C content does not increase (Virto et al., 2012). In addition, several studies show a gradual increase in SOC stocks under no-tillage, albeit differences between tillage systems become detectable only after several decades after conversion (Or et al., 2021; Angers and Eriksen-Hamel, 2008; Haddaway et al., 2017). This issue would be clarified in the discussion.

4. Even though the carbon content of both treatments is about the same, the biological soil properties seem to differ. Biodiversity in OPM is higher. This result is interesting because information on soil biology is often missing in other publications and could be part of the main text here and not be presented in an appendix. (SDG15).

Since our evaluation is mainly based on soil porosity analysis, we had included the information concerning the biological properties of the soil in an annex. But the reviewer's comment is very right. Thus, we would include a new section "Organic C storage and soil microbial diversity", just passing the information regarding biological issues from the supplementary material to the main text.

5. As bulk densities of the two treatments are about the same for 0-5 cm dept and for the 0-30 cm depth (how was the latter measured, we read only the method for 0-5 cm? Effect of high tempera) there seems to be no negative effect on soil structure by CM management which is usually assumed to take place. Big problem here is that analyses were only made to 30 cm depth and a plowpan may well form by CM management but usually occurs at 30 cm depth and deeper. Indeed, as the authors mention on line 320: deeper soil layers are needed. For a correct analysis soils should be analysed for the total rooting depth. (SDG15).

In fact, the bulk density values correspond only to the first 5 cm (0-5 cm). However, based on our field observations these values would be extrapolated up to 30 cm depth. At greater depths, a higher compaction was observed, especially in the soil under conventional tillage (CM). Nevertheless, the rest of the indicators were analyzed at 0-30 cm, as suggested by the reviewer and proposed for example by FAO for the organic C stock. All these would be clarified in the text (material and methods) and in Table 1 of our manuscript.

As already mentioned in the conclusions similar experiments at deeper soil layers are needed.

Also, in relation to SDG15, we have now considered in more detail the potential of the OPM system for erosion control, and will discuss it in the new version of the manuscript, by adding up-to-date information in this view:

"Changes in soil erosion rates depend among others on climatic conditions, land use patterns, farmers' decisions and, agri-environmental policies (Panagos et al., 2021; Mosavi et al., 2020; Grillakis et al., 2020; Eekhout and De Vente, 2020; Paroissien et al., 2015)

Recent work has shown that studies associated with soil erosion are essential, both for agricultural conservation practices and to subsidize environmental planning, where economic practices must be calculated under conservationist principles (Panagos et al., 2021; Cruz et al., 2019; Mosavi et al., 2020; Plambeck, 2020). In particular, selective application of cover crops in soil erosion hotspots, combined with limited soil disturbance measures, has been recently recommended as an effective measure for partially or totally mitigate the effect of climate change on soil losses in Europe (Panagos et al., 2021)."

**Some additional notes:**

1. **Line 45:** to study soil pores, morphological analyses are most useful, if only because different types of macropores can be distinguished: e.g. channels of roots or animals or cracks. Deriving pore sizes form moisture retention data is an indirect, approximate method. Later (lines 260, 266) micromorphology is mentioned. That should also be done upfront.

The reviewer is right, micromorphological analysis, as a (more) direct method of pore space characterization, should be mentioned in the paragraph starting on line 45. However, the use of thin sections limits the study of pores larger than the thickness of the film. This is usually 20 microns (Virto et al., 2013).

2. **Line 190:** no significant compaction? Could be deeper in the CM soil, see comment above.

This issue was treated and clarified above.

**Line 319:** infiltration at the surface is mentioned in the context of the pore analysis. But infiltration rates can be measured and this is very important for the Mediterranean environment, certainly when considering climate change where showers will become more intense. CM has more macropores, so the authors suggest that infiltration rates would be higher than in OPM with fewer macropores. But the CM soil is less stable so crusts may form rapidly, the more so since there are no cover crops and crop residues are removed. So just linking the occurrence of physically derived macropores to infiltration rates is unrealistic without measurements of such rates. In fact, the reasoning should be reversed: measure infiltration rates and then explain differences by looking at pore patterns.

This is true; it is rather speculative to directly relate infiltration to soil macroporosity. We will limit ourselves to the description of porosity, while indicating that it would be necessary to have infiltration measurements, ideally with controlled suction (e.g.,disc infiltrometer).

4. The authors conclude that more water is stored in the OPM treatment and this should be favorable for plant growth. But this is a statement based on static moisture retention measurements while storage is determined by in- and outflow from a certain soil volume, a dynamic process as is the moisture supply to plants. Numerous dynamic simulation models are available to quantify this process, that needs hydraulic conductivities of the soil. The authors seem to be unaware of modern soil physical theory.

The reviewer is right; we cannot be conclusive in this issue with our results. We could rephrase that sentence by saying that our results suggest a higher water storage capacity in OPM. However, to properly quantify water storage capacity inflow and outflow from the soil should be estimated. This issue would be clarified in the text; more precisely in the discussion (line 201) and in the conclusion (line 306).

5. As the soil index by Dexter does not provide any valuable results (lines 226, 317) it should receive much less attention and might as well be omitted.

We agree with the reviewer, we have decided that the Dexter index deserves less attention and we will not be mentioned in the objectives:

"The objective of this study was to assess the continuous application, throughout 18 years, of an innovative soil and crop management –in comparison with conventional management– for the improvement of the soil physical condition, and the optimization of the soil water balance, in rainfed cereal agrosystems in semi-arid land. This evaluation will be carried out through the analysis of detailed SWRCs and soil structure, i.e., the size-distribution of stable macro- and microaggregates and their relation to organic C storage."

It should be noted that the main strength of our work is the use of detailed SWRCs that allows a better and more reliable adjustment of any index based, precisely, on the different shape of these curves. In most of the published works in which the S-index was applied, the corresponding SWRCs were constructed from a small number (less than 10) of water content-suction values, obtained from sandbox and/or pressure plates apparatus. To evaluate to what extent the degree of detail of the SWRCs is relevant we have recalculated the van Genuchten' parameters and the S index considering only 9 water content-suction values –the most frequently used, according to the literature–. Not surprisingly, the values obtained differed markedly –with no apparent bias– from those reached using the full dataset (about 100 evenly distributed points).

Taking advantage of our continuous SWRCs and following the reviewer's #5 suggestion we have determined the water retention energy index (WRa) (Armindo and Wendroth, 2016) obtained from numerical integration including all the points of the SWRC. Needless to say, that the accuracy of this index is highly conditioned by the degree of detail of the SWRCs. Moreover, this index presents an adequate sensitivity for smaller-scale, high-precision applications and for capturing the dynamic evolution of the soil physical state (Armindo and Wendroth, 2016) (see more details in the answers to reviewer #5).

A possible new title might be: *Effects of innovative soil management on topsoil properties of a Mediterrenean soil*. This could focus on several interesting results identified above. Again, I would recommend they would frame their story in an SDG context.

As mentioned above, a new title is proposed based on the reviewer suggestion: *Effects of innovative long-term soil management on topsoil properties of a Mediterranean soil based on detailed water retention curves.*

The authors should realize that OPM is, in fact, a form of regenerative agriculture, studied in the USA. I recommend that they check with the National Soil Health Institute website. info@soilhealthinstitute.org.

Many thanks for your recommendation.

J.Bouma.

**References**

Angers, D. A. and Eriksen-Hamel, N. S.: Full-Inversion Tillage and Organic Carbon Distribution in Soil Profiles: A Meta-Analysis, Soil Sci. Soc. Am. J., 72, 1370–1374, https://doi.org/10.2136/sssaj2007.0342, 2008.

Armindo, R. A. and Wendroth, O.: Physical Soil Structure Evaluation based on Hydraulic Energy Functions, Soil Sci. Soc. Am. J., 80, 1167–1180, https://doi.org/10.2136/sssaj2016.03.0058, 2016.

Council of the European Union: Council Directive 91/676/EEC of 12 December 1991 concerning the protection of waters against pollution caused by nitrates from agricultural sources, , L 269, 1–15, 2008.

Cruz, D. C., Benayas, J. M., Ferreira, G., Monteiro, A. L., and Schwartz, G.: Evaluation of soil erosion process and conservation practices in the paragominas-pa municipality (Brazil), Tech. Geogr., 14, 14–35, https://doi.org/10.21163/GT_2019.141.02, 2019.

Eekhout, J. P. C. and De Vente, J.: How soil erosion model conceptualization affects soil loss projections under climate change, Prog. Phys. Geogr., 44, 212–232, https://doi.org/10.1177/0309133319871937, 2020.

Grillakis, M. G., Polykretis, C., and Alexakis, D. D.: Past and projected climate change impacts on rainfall erosivity: Advancing our knowledge for the eastern Mediterranean island of Crete, 193, 104625, https://doi.org/10.1016/j.catena.2020.104625, 2020.

Haddaway, N. R., Hedlund, K., Jackson, L. E., Kätterer, T., Lugato, E., Thomsen, I. K., Jørgensen, H. B., and Isberg, P. E.: How does tillage intensity affect soil organic carbon? A systematic review, BioMed Central, 1–48 pp., https://doi.org/10.1186/s13750-017-0108-9, 2017.

Mosavi, A., Sajedi-Hosseini, F., Choubin, B., Taromideh, F., Rahi, G., and Dineva, A. A.: Susceptibility mapping of soil water erosion using machine learning models, 12, 1–17, https://doi.org/10.3390/w12071995, 2020.

Or, D., Keller, T., and Schlesinger, W. H.: Natural and managed soil structure: On the fragile scaffolding for soil functioning, Soil Tillage Res., 208, 104912, https://doi.org/10.1016/j.still.2020.104912, 2021.

Panagos, P., Ballabio, C., Himics, M., Scarpa, S., Matthews, F., Bogonos, M., Poesen, J., and Borrelli, P.: Projections of soil loss by water erosion in Europe by 2050, Environ. Sci. Policy, 124, 380–392, https://doi.org/10.1016/j.envsci.2021.07.012, 2021.

Paroissien, J. B., Darboux, F., Couturier, A., Devillers, B., Mouillot, F., Raclot, D., and Le Bissonnais, Y.: A method for modeling the effects of climate and land use changes on erosion and sustainability of soil in a Mediterranean watershed (Languedoc, France), J. Environ. Manage., 150, 57–68, https://doi.org/10.1016/j.jenvman.2014.10.034, 2015.

Plambeck, N. O.: Reassessment of the potential risk of soil erosion by water on agricultural land in Germany: Setting the stage for site-appropriate decision-making in soil and water resources management, Ecol. Indic., 118, 106732, https://doi.org/10.1016/j.ecolind.2020.106732, 2020.

Virto, I., Barré, P., Burlot, A., and Chenu, C.: Carbon input differences as the main factor explaining the variability in soil organic C storage in no-tilled compared to inversion tilled agrosystems, Biogeochemistry, 108, 17–26, https://doi.org/10.1007/s10533-011-9600-4, 2012.

Virto, I., Fernández-Ugalde, O., Barré, P., Imaz, M. J., Enrique, A., Bescansa, P., and Poch, R. M.: Micromorphological analysis on the influence of the soil mineral composition on short-term aggregation in semi-arid Mediterranean soils, Spanish J. Soil Sci., 3, 116–129, https://doi.org/10.3232/SJSS.2013.V3.N2.07, 2013.

---

## Author Comment (AC3)

**General comments:**

The purpose of the manuscript is a simple comparison of the influence of two management systems, conventional cropping and direct-drilling rain-fed cropping in the physical properties of a soil in the north of the Spain. The selected physical properties are the S index of Dexter (2004) and the size distribution of the stable aggregates. Neither the objectives nor the methods represent a new contribution to the fields of Agronomy or Soil Science.

Nevertheless, the relevance of reduced tillage systems for the Mediterranean countries, and the long duration of the field trials, deserve an opportunity for the authors after a thorough revision of the manuscript. Some recent contributions as, for instance, Or et al. (2021) could be inspiring for such a revision.

First of all, thank you very much for the helpful comments and suggestions you have made about our manuscript.

By re-reading our own manuscript after the reviewer comments, we realize that we did not properly clarify certain important points necessary to better understand the relevance that we sincerely believe our work has.

We evaluate more than just a direct-drilling but an innovative soil and crop management system (OPM) in Navarre (Mediterranean conditions). Moreover, there is to our knowledge, no agricultural field in all of Navarre where soil and crop management as proposed herein is practiced and even less for almost two decades, with the exception of our relatively small area where OPM was introduced by a pioneer farmer nearly 20 years ago.

As the reviewer is well aware, the crop performance under no-till is strongly dependent on the crop type and climate (Or et al., 2021) and also soil type. Then, no-till may not be suitable for all conditions (Pittelkow et al., 2015). In fact, conventional tillage -in carefully managed agricultural soils- may be imposed when no-tillage would lead to chronic and unacceptable yield losses. (Or et al., 2021); in fact, no-till often results in reduction in crop yields of ca. 10 % in some areas (Or et al., 2021). Although we do not have precise data on crop yields in both treatments (CM vs OPM) some rough data are available from the farmers managing the fields (see Table 0 below). From the data provided by these farmers we cannot assure that the OPM treatment was better than the CM in terms of yield. In fact, there is usually no difference between the two treatments because the limiting factor is mostly water. Even so, the data suggest that farmers will not suffer a decrease in the yields of their fields by implementing sustainable practices, i.e. OPM.

Table 0. Average crop yield (2016-2021) of OPM and conventional agricultural fields under conventional tillage (CM), as reported by farmers.

| Crop | Yields (t/ha) | |
| --- | --- | --- |
| | OPM | CM |
| Wheat | 6.8 - 9.3 | 5.5 - 7.0 |
| Barley | 5.8 - 8.0 | 5.0 - 6.5 |
| Rapessed | 2.0 - 4.0 | 2.0 - 3.0 |
| Legumes | 2.2 - 3.5 | 1.7 - 2.5 |

The uniqueness of our pioneer OPM area, and then the relevance of our work should be better and clearly pointed out in the text, especially in the Introduction and in section 2.1. (Study site [former 'zone'] and treatments).

On the other hand, another key issue in our work is the use of detailed SWRCs that allows a better and more reliable adjustment of any index based, precisely, on the different shape of these curves. In most of the published works in which the S-index was applied, the corresponding SWRCs were constructed from a small number (less than 10) of water content-suction values, obtained from sandbox and/or pressure plates apparatus. To evaluate to what extent the degree of detail of the SWRCs is relevant we have recalculated the van Genuchten' parameters and the S index considering only 9 water content-suction values –the most frequently used, according to the literature–. Not surprisingly, the values obtained differed markedly –with no apparent bias– from those reached using the full dataset (about 100 evenly distributed points).

Taking advantage of our continuous SWRCs and following the reviewer's #5 suggestions we have determined the water retention energy index (WRa) (Armindo and Wendroth, 2016) obtained from numerical integration including all the points of the SWRC. Needless to say, that the accuracy of this index is highly conditioned by the degree of detail of the SWRCs. Moreover, this index presents an adequate sensitivity for smaller-scale, high-precision applications and for capturing the dynamic evolution of the soil physical state (Armindo and Wendroth, 2016) (see more details in the answers to reviewer #5).

The Introduction section does not contain a comprehensive, updated perspective of the conservation agriculture and the quality of the soil. Consequently, the objectives, (lines 77-82) are very imprecise.

Following the reviewer's advice, we have made a revision and have drafted the following paragraph to be included in the introduction:

"Conservation agriculture (CA), and other soil management strategies implying a reduction of tillage have been reported to reduce soil degradation in different agroecological situations (Verhulst et al., 2010; Sartori et al., 2022), and in some cases are designed for this purpose (Virto et al., 2015).

The reasons reported for its adoption in Europe are several. In Northern Europe soil erosion control, soil crusting in loamy soils and the need to increase soil organic C storage, as well as soil trafficability are widely cited as reasons for CA implementation (Lahmar et al., 2007). In the Mediterranean countries, soil water storage and water-use efficiency can be added to this list of reasons (De Turdonnet et al., 2007). However, different studies show that the effectiveness of CA in solving these problems can be site-dependent (Costantini et al., 2020; Chenu et al., 2019). In fact, the most widely reported benefits of CA in Southwestern Europe in relation to erosion are the increased soil infiltrability and/or the protective effect of crop residues on the soil surface (Gómez et al., 2009; Espejo-Pérez et al., 2013; Virto et al., 2015), although this seems to be related to the type of soil and to the presence and activity of earthworms. In Spain, the soil water-retention capacity has been observed to be greater in semi-arid land under no-tillage (Fernández-Ugalde et al., 2009; Bescansa et al., 2006).

Other positive effects of CA on soil quality observed in semi-arid rainfed agricultural systems in Spain are related to soil organic C and nutrients storage (Ordóñez Fernández et al., 2007)".

The Material and methods section is rather incomplete, with some inaccuracies that will be commented later. The climate properties of the study zone are missing. The description of the soil is limited the mention of the subgroup in the Soil Taxonomy scheme, the textural class of the upper soil horizon and Tables 1 and A1. No explanation is given in the text of the methods followed for the determination of the data of Table1. However, the details of the soil water retention in subsection 2.2 and of the aggregate size fractionation in subsection 2.4 are excessive including Figure 1.

The reviewer is very right, some of the soil information -clay, silt and sand contents- was originally included in the supplementary material of the manuscript. However, since this is relevant information, it would be included in Table 1 along with the methods followed for the data determination. In addition, at the suggestion of reviewer #4, a statistical analysis of the parameters in Table 1 will be performed and those showing significant differences ($p < 0.05$) between treatments will be marked in bold. This can be summarized in a paragraph that would be included in the new version, as follows:

"The physical-chemical analysis of the soils shown in Table 1 was done using standard methods. In particular, soil pH was analyzed in a 1:2.5 soil:water solution as in Hendershot and Lalande (1993), organic C content by wet combustion as in Tiessen and Moir, (1993), carbonates were determined in a modified Bernard's calcimeter following Pansu and Gautheyrou (2003a), and the electrical conductivity in a soil:water solution similar to that for pH analysys (Pansu and Gautheyrou, 2003b). The soil texture was determined by the pipette method. All analyses were conducted on air-dried samples ground to 2 mm. Finally, the bulk density was determined using the Hyprop device (see below) from undisturbed samples extracted from the first 5 cm of the soil profile.

Table 1. Physical-chemical properties of the topsoil (0-30 cm) in OPM and CM treatments and the textural characterization of both treatments. Mean ± standard deviation of the mean (n = 3). Statistically significant differences ($p < 0.05$) are in bold."

| Treatment | Optimized (OPM) | Conventional (CM) |
|---|---|---|
| Bulk density (0-5 cm) (g·cm$^{-3}$) | 1.26 ± 0.05 | 1.26 ± 0.15 |
| pH | 8.00 ± 0.05 | 8.01 ± 0.01 |
| Organic C (%) | **1.80 ± 0.10** | **1.51 ± 0.14** |
| CE (µS·cm$^{-1}$) | **483 ± 5.66** | **795 ± 4.24** |
| Carbonates (%) | **31.6 ± 0.09** | **32.5 ± 0.14** |
| Sand (Coarse) (%) | 5.05 ± 0.08 | 5.79 ± 0.33 |
| Sand (Fine) (%) | 30.9 ± 1.00 | 31.7 ± 1.25 |
| Silt (%) | 47.2 ± 1.23 | 43.7 ± 0.93 |
| Clay (%) | **16.9 ± 0.46** | **18.5 ± 0.46** |
| Texture (USDA) | *Loam* | *Loam* |

On the other hand, as indicated by the reviewer, a brief description of the climatic properties of the study area will be included in the Material and Methods section:

"This is an area with a dry temperate Mediterranean climate, according to Papadakis (1967). The mean annual precipitation is 550 mm year$^{-1}$, and the Thornthwaite mean annual evapotranspiration is 711 mm year$^{-1}$ (Gobierno de Navarra Meteorología y Climatología de Navarra, 2022)."

The Results and discussion section is incomplete as well. The authors have chosen the van Genuchten soil water retention equation, but, in addition to the absence of the fitted values of its parameters in the text, one misses some consideration of other alternatives as, for instance, the multimodal equations, which according to Jensen et al. (2019), can reflect better the effects of management systems in the soil properties. I have expected some changes in properties like the bulk density, and the presence of some surface crusts, but, apparently, they were not found. The Figure 3 could not give an adequate information on the size distribution of the pores since it is estimated from the soil water retention curve through equation (2). The discussion of the results is fragmentary.

Following the reviewer suggestions we plotted our experimental results as differential functions [dΘ/d(log h) vs log h(h)] seeking for a multimodal behavior: all the curves analyzed were of the uni-modal type. But it should be noted that suction values do not exceed 150 kPa and according to Dexter et al. (2008) (cf. their Fig. 3) and Jensen et al. (2019) (cf. their Fig. 2) findings the second peak defining a bimodal behavior seems to appear at suction around 1000 kPa. Then, we tried again incorporating to the dataset the water content-suction measurements at 1500 kPa (obtained using pressure plates) with the same result, i.e. unimodal behavior. But this could be an artifact of the dataset since there is a wide experimental gap between 150 kPa and 1500 kPa, i.e. no measurements in between.

Despite this, we tried the double-exponential equation for soil water retention proposed by Dexter et al. (2008) (Eq.1) with our experimental results:

$$\theta = C + A_1 e^{\left(-\frac{h}{h_1}\right)} + A_2 e^{\left(-\frac{h}{h_2}\right)} \tag{1}$$

Where $\Theta$ is the gravimetric water content; $C$ is the residual water content (asymptote of the equation); the amount of matrix and structural pore space are proportional to $A_1$ and $A_2$, respectively. The values of $h_1$ and $h_2$ are the characteristic pore water suctions at which the matrix and structural pore spaces empty, respectively (Dexter et al., 2008).

Table 2 shows the values of the fitted parameters of equation (1) using our experimental dataset. Unexpected at first sight, the structural pore space would have been reduced by 55 % as a result of no-tillage (OPM) (cf. $A_2$ values, Table 2), while the matrix pore space values remain rather constant in both treatments (see $A_1$ values in Table 2). This could be explained by an increase in soil fragments rather than soil aggregates in the CM treatment in comparison with OPM treatment. Soil aggregates and fragments may look similar but are formed by different processes and have different properties (Or et al., 2021): soil fragments form by mechanical forces of tillage; they tend to be mechanically weak and coalesce upon wetting with macroposity collapsing within a single season. Instead, soil aggregation is stimulated by biological activity with biopolymers and hyphae that stabilize and bind soil particles.

Table 2. Average values of the fitted parameters of the double-exponential water retention equation by Dexter et al. (2008) obtained with the experimental dataset.

| Treatment | Parameters of the Dex model | | | | | |
|---|---|---|---|---|---|---|
| | C | $A_2$ | $h_2$ | $A_1$ | $h_1$ | RMSE |
| | $m^3 \cdot m^{-3}$ | $m^3 \cdot m^{-3}$ | hPa | $m^3 \cdot m^{-3}$ | hPa | $m^3 \cdot m^{-3}$ |
| OPM | 0.25 | 0.06 | 29.94 | 0.11 | 865.08 | 0.005 |
| CM | 0.2 | 0.10 | 25.16 | 0.11 | 737.43 | 0.003 |

All this is consistent with our previous results as described next. The fast desorption rate at low suction observed in CM compared with OPM (cf. Fig 2 in our manuscript) could be the result of macropores due to large soil fragments induced by tillage. However, more stable (biologically-formed) macroaggregates were observed in the OPM treatment compared with CM (cf. Fig 4 in our manuscript).

But all the above should be taken with caution since, as aforementioned, our SWRCs do not show a bi-modal behavior and then the use the Dexter's et al. (2008) equation -and what is inferred therefrom-could be questioned.

Regarding Figure 3, to avoid misinterpretation, it should be clarified in its caption that pore sizes are estimated from equation (2).

**Specific comments**

Line 28: what is 'the studied depth'?

Thanks for showing this ambiguity: the studied depth is the first 30 cm of the soil profile. This sentence should be reformulated.

Lines 51-53: The definition of the soil water retention is very imprecise. Why soil water retention 'is mainly associated with the porous system' 'at low suctions'?

Sorry, that sentence is indeed ambiguous, it should instead read:

"Soil water retention curve (SWRC) is the relationship between soil water matric potential and soil water content. Since water suction/retention is largely determined by pore sizes SWRCs are a valuable tool to diagnose the physical condition of soils (Dexter, 2004a, b; Pires et al., 2017)."

Line 56: The treatment of the air entry state is, again, imprecise. The term 'so-called' is unnecessary.

We agree, the term "so-called" would be eliminated.

Lines 67-70: This paragraph is unclear.

Sorry, it is indeed unclear, it should be reformulated as follows:

"The inflection point can be determined directly by hand from the SWRC if there are enough accurate measurement points (Dexter, 2004a). Alternatively, it would be more appropriate to fit the SWRC to a mathematical function and then to calculate the slope at the inflection point in terms of the parameters of the function. To do this, one of the best known functions is that proposed by van Genuchten (1980) for which, in turn, pedo-transfer functions are available for estimation of its parameters (Dexter, 2004a)."

Lines 71 and 77: The two sentences are almost repeated.

This is true, thanks. The objective should be rewritten as the reviewer suggests; including also a modification proposed by the reviewer #1, i.e., do not highlight the use of the Dexter's index:

"The objective of this study was to assess the continuous application, throughout 18 years, of an innovative soil and crop management –in comparison with conventional management– for the improvement of the soil physical condition, and the optimization of the soil water balance, in rainfed cereal agrosystems in semi-arid land. This evaluation will be carried out through the analysis of detailed SWRCs and soil structure, i.e., the size-distribution of stable macro- and microaggregates and their relation to organic C storage."

Lines 88-90: A more complete description of the soils should have been very helpful to understand their behavior. One cannot trust 'the visual inspection' to affirm that the soil is homogeneous. As in the line 28, could 'the study depth' be defined?

As mentioned above, Table 1 giving the main properties of the soil would include more information previously included in the supplement materials.

Regarding the low reliability of a visual inspection, in fact, for visual inspection we actually mean field soil description following standard procedures (i.e., penetration resistance, soil structure, consistence, etc). Anyway, to give less relevance to the in situ soil description this sentence could be reformulated as follows:

"The physical-chemical analysis of soils (Table 1) showed high homogeneity of the soil at the study depth (0-30 cm) regarding the most relevant physical-chemical properties related to moisture retention, except for the content of organic C (which can be related to the change in management). In addition, in situ standard soil description corroborated the homogeneity of the topsoil (0-30 cm)."

Line 105: The term 'coverer' is very odd.

The sentence has been reformulated to avoid the use of the term "coverer", and to provide more accuracy about the fertilization timing:

"In both treatments, mineral fertilization consisted of phosphorus addition before seeding (120-150 kg·ha$^{-1}$ of triple superphosphate 0-46-0) and nitrogen supply of 180 kg N·ha$^{-1}$ (split and distributed into two cover dressings at 60 kg N·ha$^{-1}$ and 120 kg N·ha$^{-1}$ in January and March, respectively) as urea."

Table 1: The evaluation method must be indicated in the text and the relevant details like the soil-water ratio for the suspension in the measurement of the pH and of the electrical conductivity.

As mentioned above, Table 1 was completed including this piece of information.

Lines 132-133: To establish the relationship between the gravimetric and volumetric water contents, one only need to know the bulk density of the soil. The sentence is confusing.

The sentence is indeed confusing, sorry. It would be reformulated as follows:

"Gravimetric water content can be expressed as volumetric content since bulk density is known"

Lines 142-145: I could not find equation (1) in the article of Dexter (2004a). In fact, I cannot find the relation of this equation with the van Genuchten (1980) equation.

It is true that the phrase is a bit confusing. It will be clarified in the text which is equation (1) and which is equation (2):

The value of S was calculated in two different ways: i) from a sigmoidal function fitted to experimental data (Eq. 1), and ii) from the adjusted parameters of the van Genuchten (1980) function (Eq. 2) (Dexter, 2004a):

$$y = \frac{a}{1 + e^{-(\frac{x - x_0}{b})}} \tag{1}$$

Where $y$ is the logarithm of suction (hPa), $x$ is the gravimetric moisture (kg·kg$^{-1}$), and $a$, $b$, $x_0$ are parameters of the equation

$$\theta_h = (\theta_{sat} - \theta_{res})[1 + (\alpha h)^n]^{-m} + \theta_{res} \tag{2}$$

Where $h$ is the soil matric potential (hPa), $\theta_h$ (m$^3$·m$^{-3}$) is the measured soil water content at matric potential $h$, $\theta_{res}$ is the residual water content (m$^3$·m$^{-3}$), $\theta_{sat}$ is the saturated water content (m$^3$·m$^{-3}$), $\alpha$ (hPa$^{-1}$), $n$ (-) and $m = 1 - (1/n)$ (-) are the van Genuchten parameters.

Lines 150-154: The equation was not proposed by Jurin (1718) as one can check reading such an article. Jurin described his observations in that contribution, but the equation was later formulated by Young and Laplace. This equation is usually known as the Young-Laplace equation (e.g. Adamson, 1967, § I.10).

Thank you very much for the clarification.

Line 154: Is the (soil water) potential is mentioned, the sign should be negative. The term 'suction' is more appropriated here if the symbol 'h' has been used for the 'height' in the line 152.

The reviewer is right. We will replace the term "potential" with "suction".

Line 195: The term 'water capacity' was already defined, at least, by Arnold Klute in 1952.

Thank you very much for the clarification.

Line 196: How 'field capacity' is defined in the text? This term must be precisely defined.

The moisture content corresponding to the inflection point of the hydraulic conductivity vs. moisture content curve -provided by the Hyprop device- would correspond to the field capacity. This should be better defined in the text as the reviewer indicates.

Table 2: as indicated above the soil water potential is negative.

In fact, soil suctions (positive values) are mentioned in Table 2 and not water potential.

Lines 231-233: If the authors are using a structural index, they should not compare the texture but the structure of the soil.

What we are trying to explain in this sentence is that the inflection point of the SWRCs in both treatments would not be affected by the texture since this does not change between treatments. For instance, the S value tends to decrease with increasing clay content (Dexter et al., 2004).

Line 306: The term 'capillary/available water' is misleading.

We agree with the reviewer. The term "capillarity" will be deleted and "available water" will be maintained.

Line 322: I do not think that tillage should be 'suppressed'. The natural consolidation of the soil surface might be alleviated by occasional shallow tillage operations.

The reviewer is right, the term "suppressed" should be replaced by "minimum tillage" since although up to now the OPM plot has not been ploughed, it may be necessary to eventually do that to avoid excessive compaction.

Finally, we agree with the following group of suggestions proposed by the reviewer and will include them in the text:

**Specific comments**

Line 18: instead of 'unit' it must be written subgroup.

Lines 25 and 26: 'significant differences' must be replaced by a statistic parameter.

Lines 33-38: The paragraph should be rewritten to improve its comprehension.

Lines 45-47: The sentence is very similar to that of the lines 37-38.

Line 85: for the sake of precision write 'subgroup' instead 'type' and mention the Soil Taxonomy.

Line 86: The textural class should be silt loam, according to the particle size information of Table A1, where it was correctly indicated.

Line 129: Use 'matric component of soil water potential' instead of 'hydric potential' to be more precise.

**Technical corrections:**

Line 54: Writing 'saturated' soil is enough.

Lines 146-147: The use of the acronym 'SUHC' is needless.

Lines 160 and 364: The correct name is Elliott.

Line 280: The Table reference is A1, not 'S1'.

Line 235 and caption of Figure 3: The proper adjective is logarithmic.

Incomplete references: Lines 366, 369-370, 373-374, 381, 387-388, 391, 407, 419.

The reference of lines 441-442 is not mentioned in the text.

**References:**

[revised manuscript text omitted]

---

## Author Comment (AC5)

The authors aiming for comparison of different regenerative soil management systems in comparison to conventional managed systems. They use various measures of soil structure and water retention to evaluate the quality of the management options. The topic is highly relevant for the adoption of cropping systems to climate change.

Unfortunately, the quality of the study does not convince me to suggest a publication in SOIL. The study is very weak in terms of field replications and study sites that leading in total to 6 samples. The analytical tools are basic and provide no innovative approaches. For such a small sample set one could expect much deeper analytical afford. Method descriptions partially missing and some data I found by luck in the supplements. In the cause of the review process I stopped marking down all individual specific comments. There was simply too much to correct and my time is limited.

**General comments**

The introduction should summarize the state of the art and introduce to the relevance of the topic. Unfortunately a larger part of the introduction (L51-70) contained technical information and methodological information. I recommend to go deeper into literature on soil management options of arable land and the connection between management of soil structure and water budget.

The study site is not characterized well. Climate data are missing at all and the distance between the two study sites is unclear. A map would help. No information on the type of management (experimental field trial or on farm research) is provided. Since this are calcareous soils, information about parent material would be helpful. How deep below soil surface starts the bedrock? Soil texture should be measured in replicates. The method must be described. Please show the data in the main manuscript from each sample. There must be a prove (correlation etc.) that there are no texture based differences on soil properties such as soil OC or CEC. It is recommended to show values from deeper soil layers (>30cm). Otherwise the effect of texture should be evaluated. There are also several uncertainties on fertilizer management: What form of OM amendments, how much, when in the crop rotation? What kind of cover crops? Please provided more details to the sampling design: Size of the fields, distance between the sampling points in a map.

Unsing an ANOVA approach for statistic comparison requires the assumption of normality. Further, the sample size of 3 is very low and likely not the suitable measure. I recommend using a simple T-test, depending on the distribution of the data.

The data set on microbial parameters should be incorporated in the main manuscript. Methods must descried properly. The same is true for the OC measurements. Have the carbonates be removed from the samples?

In the conclusion there something written with vegetable cover, that was not discussed before. I did not get this point. The author's proclaimed the "optimized managtemen" practices for the whole Mediterranean region. From this very limited data set at one sampling site it is not possible to scale up the management tools across the whole Mediterranean environment. Also the effects in the subsoil have not been taken into account.

I also highly recommend a professional language check. Many basic rules for preparing a scientific publication are ignored. For example: The manuscript is overloaded with double brackets, grammar errors or punctuation errors. Many paragraphs are not accessible, even after reading several times.

**Specific comments:**

L39 change eliminate to avoid. Further, cover cropping have nothing to do with soil tillage practices, examples for reduced soil tillage are e.g. mini tillage (0-10cm) , Cultivator application and everything that avoids to invert the soil of 0-30cm.

L43 the annual soil water balance primarily depend on precipitation: I suggest optimisation of the infiltration and water storing capacity

L51-70 this belongs to the materials &method section. If you write a paper on the methods you could bring this here.

L78 managed

L86-88 the sentence is overloaded with brackets. Avoid double brackets and remove some them. Keep this in mind for the rest of the manuscript.

L97-102 this is unclear. Is only the cereal straw removed from the fields? I never heard that straw of legumes and rapeseed is removed?

L107 which form of OM amendments, how much nutrients and OC therein?

L245 - following. Use space between ± and the number. The same is true for >. Only between the numbers an % there should not be a space.

L235-238 no consistency Fig. Figure. The two paragraphs double.

L246 texture homogeneity was not measured.

L311 where does the vegetable cover comes from?

We sincerely appreciate your thorough review and your helpful comments and suggestions that significantly helped to improve our manuscript. We send you a detailed response to each of your comments, hoping that you reconsider your decision. In any case, many thanks for your time and effort.

As the reviewer's comments/suggestions dealt with in different paragraphs, we have grouped them by topics, and headed with key words (in bold) referring to the main subject under discussion:

***Research scope***. ***Study site characterization.*** We believe it is appropriate to begin first providing clarifications regarding the experimental site in order to, in turn, better understand other aspects of the work, such as sampling and repetitions, and also to emphasize the relevance and scope of our research.

We agree with the reviewer that our findings are too narrow to be a comprehensive contribution to the sustainability of Mediterranean agrosystems considering all the aspect that sustainability development really involved, as clearly remarks the reviewer. In fact, our main objective is the evaluation of an innovative -and pioneer in our region (Navarre)- soil and crops management strategy on topsoil properties. Indeed, to our knowledge, there is no other agricultural field in the whole region of Navarre where soil and crop management as proposed herein (OPM) is practiced and even less for almost two decades, with the exception of our small (2 hectares) OPM test area were a pioneer farmer works for almost 20 years (farm research). This represents, in our view, the uniqueness of our OPM study. This relevance of our work would be better and clearly pointed out in the text, especially in the Introduction and in section 2.1. (Study site [former 'zone'] and treatments), when a new version is produced after the Editor's decision.

In this sense, we believe that the actual scope of our work could be unambiguously presented as follows. First, deleting the questioned statement that "the study contributes to higher sustainability of Mediterranean agrosystems". Second, reformulating the title following the reviewer's suggestion: *Effects of innovative long-term soil management on topsoil properties of a Mediterranean calcareous soil based on detailed water retention curves.*

***State of the art of conservation agriculture***. Following the reviewer's advice, we have made a revision and have drafted the following paragraph to be included in the introduction:

"Conservation agriculture (CA), and other soil management strategies implying a reduction of tillage have been reported to reduce soil degradation in different agroecological situations (Verhulst et al., 2010; Sartori et al., 2022), and in some cases are designed for this purpose (Virto et al., 2015).

The reasons reported for its adoption in Europe are several. In Northern Europe soil erosion control, soil crusting in loamy soils and the need to increase soil organic C storage, as well as soil trafficability are widely cited as reasons for CA implementation (Lahmar et al., 2007). In the Mediterranean countries, soil water storage and water-use efficiency can be added to this list of reasons (De Turdonnet et al., 2007). However, different studies show that the effectiveness of CA in solving these problems can be site-dependent (Costantini et al., 2020; Chenu et al., 2019). In fact, the most widely reported benefits of CA in Southwestern Europe in relation to erosion are the increased soil infiltrability and/or the protective effect of crop residues on the soil surface (Gómez et al., 2009; Espejo-Pérez et al., 2013; Virto et al., 2015), although this seems to be related to the type of soil and to the presence and activity of earthworms. In Spain, the soil water-retention capacity has been observed to be greater in semi-arid land under no-tillage (Fernández-Ugalde et al., 2009; Bescansa et al., 2006).

Other positive effects of CA on soil quality observed in semi-arid rainfed agricultural systems in Spain are related to soil organic C and nutrients storage (Ordóñez Fernández et al., 2007)".

***Experimental design and sampling***. First, it should be clarified that the OPM area is close to the CM area, and the whole surface accounts for around three hectares. In relation to the comparability of the two practices, this means that fields are close enough to consider management as the only relevant factor of change. Three composite samples –including three sub-samples each– for each treatment (OPM and CM) were taken. Both composite samples and subsamples were randomly taken within the whole studied area.

Regarding sampling depth, the bulk density values reported correspond only to the first 5 cm (0-5 cm). However, based on our field observations these values would be extrapolated up to 30 cm depth. Nevertheless, the rest of the indicators were analyzed at 0-30 cm. All these would be clarified in the text (material and methods) and in Table 1 (see below). As already mentioned in the conclusions, similar experiments at deeper soil layers are needed.

***Climate and soil characterization***. The reviewer is very right, some of the soil information -clay, silt and sand contents- was originally included in the supplementary material of the manuscript. However, since this is relevant information, it would be included in Table 1 along with the methods followed for the data determination, which did not include decarbonation of the soil samples. In addition, a statistical analysis of the parameters in Table 1 will be performed and those showing significant differences ($p < 0.05$) between treatments will be marked in bold. This can be summarized in a paragraph that would be included in the new version, as follows:

"The physical-chemical analysis of the soils shown in Table 1 was done using standard methods. In particular, soil pH was analyzed in a 1:2.5 soil:water solution as in Hendershot and Lalande (1993), organic C content by wet combustion as in Tiessen and Moir, (1993), carbonates were determined in a modified Bernard's calcimeter following Pansu and Gautheyrou (2003a), and the electrical conductivity in a soil:water solution similar to that for pH analysys (Pansu and Gautheyrou, 2003b). The soil texture was determined by the pipette method. All analyses were conducted on air-dried samples ground to 2 mm. Finally, the bulk density was determined using the Hyprop device (see below) from undisturbed samples extracted from the first 5 cm of the soil profile.

Table 1. Physical-chemical properties of the topsoil (0-30 cm) in OPM and CM treatments and the textural characterization of both treatments. Mean ± standard deviation of the mean (n = 3). Statistically significant differences ($p < 0.05$) are in bold."

| Treatment | Optimized (OPM) | Conventional (CM) |
|---|---|---|
| Bulk density (0-5 cm) (g·cm$^{-3}$) | 1.26 ± 0.05 | 1.26 ± 0.15 |
| pH | 8.00 ± 0.05 | 8.01 ± 0.01 |
| Organic C (%) | **1.80 ± 0.10** | **1.51 ± 0.14** |
| CE (µS·cm$^{-1}$) | **483 ± 5.66** | **795 ± 4.24** |
| Carbonates (%) | **31.6 ± 0.09** | **32.5 ± 0.14** |
| Sand (Coarse) (%) | 5.05 ± 0.08 | 5.79 ± 0.33 |
| Sand (Fine) (%) | 30.9 ± 1.00 | 31.7 ± 1.25 |
| Silt (%) | 47.2 ± 1.23 | 43.7 ± 0.93 |
| Clay (%) | **16.9 ± 0.46** | **18.5 ± 0.46** |
| **Texture (USDA)** | *Loam* | *Loam* |

In relation to soil characteristics, attention was paid to sample on the areas corresponding to the same soil unit, on the alluvial plain of the Cidacos River. This soil unit contains deep soils, and the parent materials are quaternary deposits rich in calcareous rocks.

On the other hand, as indicated by the reviewer, a brief description of the climatic properties of the study area will be included in the Material and Methods section:

"This is an area with a dry temperate Mediterranean climate, according to Papadakis (1967). The mean annual precipitation is 550 mm year$^{-1}$, and the Thornthwaite mean annual evapotranspiration is 711 mm year$^{-1}$ (Gobierno de Navarra Meteorología y Climatología de Navarra, 2022)."

*Fertilizer management.* More information about fertilization should be added to the text, and can be explained as follows: "In both treatments, mineral fertilization consisted of phosphorus addition before seeding (120-150 kg·ha$^{-1}$ of triple superphosphate 0-46-0) and nitrogen supply of 180 kg N·ha$^{-1}$ (split and distributed into two cover dressings at 60 kg N·ha$^{-1}$ and 120 kg N·ha$^{-1}$ in January and March, respectively) as urea. Organic fertilization was not used in any of the study treatments until 2021, in which an organic amendment was applied to the soil without disturbing the surface in the OPM treatment. After harvest, pig slurry was applied with an average concentration of 2.5 kg N·m$^{-3}$, by means of a tanker equipped with a system of hanging pipes that deposit the product a few centimeters above the ground and at a time close to a forecasted rainfall event. The application rate was 60 m$^{3}$·ha$^{-1}$ of slurry. These rates are within the legal limits established by legislation for groundwater protection against pollution caused by nitrates from agricultural sources (EU Directive 91/676 (Council of the European Union, 2008)), as the area is within a vulnerable watershed according to this Directive."

***Cover crops.*** Description of the cover crops should be added to the text, and can be explained as follows: "In OPM, both grain and straw were also removed in the 11 first years of implementation, and only stubble remained on the surface of soil when direct seeding was implemented with minimal soil perturbation. Since then, and for the 7 remaining years, the procedure was slightly modified, and only grain was removed at harvest. Therefore, chopped straw and stubble remained on the surface of the soil before direct seeding with no disruption of the soil surface. At the same time, cover crops were introduced in the system thought it is a risky practice in rainfed Mediterranean agrosystem characterized by warm and dry summers. As such, summer cover was routinely granted in this system by letting spontaneous vegetation grow in the summer, after harvest. This vegetation was dried with herbicides before seeding the cash crops in the Fall. Also, only one year the winter crop used was *Vicia villosa* Roth, and served as a cover crop for sorghum (*Sorghum vulgare* L.), which was successfully grown in the spring-fall season despite the limiting water availability in the area."

***Biological properties information.*** Since our evaluation is mainly based on soil porosity analysis, we had included the information concerning the biological properties of the soil in an annex. But, the reviewer's comment is very right. Thus, we would include a new section "Organic C storage and soil microbial diversity", just passing the information regarding biological issues from the supplementary material to the main text.

***Analytical tools.*** We improved the discussion incorporating two different approaches.

First, we plotted our experimental results as differential functions [dΘ/d(log h) vs log h(h] seeking for a multimodal behavior: all the curves analyzed were of the uni-modal type. But it should be noted that suction values do not exceed 150 kPa and according to Dexter et al. (2008) (cf. their Fig. 3) and Jensen et al. (2019) (cf. their Fig. 2) findings the second peak defining a bimodal behavior seems to appear at suction around 1000 kPa. Then, we tried again incorporating to the dataset the water content-suction measurements at 1500 kPa (obtained using pressure plates apparatus) with the same result, i.e. unimodal behavior. But this could be an artifact of the dataset since there is a wide experimental gap between 150 kPa and 1500 kPa, i.e. no measurements in between.

Despite this, we tried the double-exponential equation for soil water retention proposed by Dexter et al. (2008) (Eq.1) with our experimental results:

$$\theta = C + A_1 e^{\left(-\frac{h}{h_1}\right)} + A_2 e^{\left(-\frac{h}{h_2}\right)} \tag{1}$$

Where $\Theta$ is the gravimetric water content; $C$ is the residual water content (asymptote of the equation); the amount of matrix and structural pore space are proportional to $A_1$ and $A_2$, respectively. The values of $h_1$ and $h_2$ are the characteristic pore water suctions at which the matrix and structural pore spaces empty, respectively (Dexter et al., 2008).

Table 2 shows the values of the fitted parameters of equation (Eq.1) using our experimental dataset. Unexpected at first sight, the structural pore space would have been reduced by 55 % as a result of no-tillage (OPM) (cf. $A_2$ values, Table 2), while the matrix pore space values remain rather constant in both treatments (see $A_1$ values in Table 2). This could be explained by an increase in soil fragments rather than soil aggregates in the CM treatment in comparison with OPM treatment. Soil aggregates and fragments may look similar but are formed by different processes and have different properties (Or et al., 2021): soil fragments form by mechanical forces of tillage; they tend to be mechanically weak and coalesce upon wetting with macroposity collapsing within a single season. Instead, soil aggregation is stimulated by biological activity with biopolymers and hyphae that stabilize and bind soil particles.

Table 2. Average values of the fitted parameters of the double-exponential water retention equation by Dexter et al. (2008) obtained with the experimental dataset.

| Treatment | Parameters of the Dex model | | | | | |
|---|---|---|---|---|---|---|
| | C | $A_2$ | $h_2$ | $A_1$ | $h_1$ | RMSE |
| | $m^3 \cdot m^{-3}$ | $m^3 \cdot m^{-3}$ | hPa | $m^3 \cdot m^{-3}$ | hPa | $m^3 \cdot m^{-3}$ |
| OPM | 0.25 | 0.06 | 29.94 | 0.11 | 865.08 | 0.005 |
| CM | 0.2 | 0.10 | 25.16 | 0.11 | 737.43 | 0.003 |

All this is consistent with our previous results as described next. The fast desorption rate at low suction observed in CM compared with OPM (cf. Fig 2 in our manuscript) could be the result of macropores due to large soil fragments induced by tillage. However, more stable (biologically-formed) macro-aggregates were observed in the OPM treatment compared with CM (cf. Fig 4 in our manuscript).

But all the above should be taken with caution since, as aforementioned, our SWRCs do not show a bi-modal behavior and then the use the Dexter's et al. (2008) equation -and what is inferred therefrom- could be questioned.

Second, taking advantage of our continuous SWRCs we have determined the water retention energy index (WRa) (Eq.2) (Armindo and Wendroth, 2016) obtained from numerical integration including all the points of each SWRC. Needless to say, the accuracy of this index is highly conditioned by the degree of detail of the SWRCs.

$$WR_a = \int_{\theta_{pwp}}^{\theta_{fc}} h(\theta)\, d\vartheta \qquad (2)$$

Where $\Theta fc$ and $\Theta pwp$ is the volumetric water content at field capacity and permanent wilting point, respectively; $h$ is suction (kPa)

WRa quantifies the total absolute energy that has to be applied by the soil to hold water in its pores between field capacity ($\Theta fc$) –i.e., after the water drainage process becomes negligible– and wilting point ($\Theta pwp$) or any moisture point $\Theta j$, where $\Theta pwp \leq \Theta j < \Theta fc$.

This index presents an adequate sensitivity for smaller-scale, high-precision applications and for capturing the dynamic evolution of the soil physical state (Armindo and Wendroth, 2016). More precisely, in the case of two SWRCs measured before and after some natural or anthropogenic changes (e.g., tillage), these energy indices can be used to quantify the change in soil physical quality status (Armindo and Wendroth, 2016). For instance, Fuentes-Guevara et al. (2022) found significantly correlation between hydraulic-energy based indices –including WRa– with some physical properties before and after land leveling operations, indicating their capacity to capture soil structure changes.

We have determined the index for the suction range between field capacity (ca. 10 kPa) and a moisture content corresponding to ca.150 kPa (maximum operating value of the Hyprop device) (Table 3).

Table 3. Absolute water retention energy (WRa) values for the two treatments (OPM, CM).

| Treatment | WRa (kPa) |
|---|---|
| OPM - repetition 1 | 4.8 |
| OPM - repetition 2 | 4.9 |
| OPM - repetition 3 | 4.1 |
| CM - repetition 1 | *5.4* |
| CM - repetition 2 | 3.5 |
| CM - repetition 3 | 3.3 |

The soil under OPM (WRa= 4.6±0.5) seems to have better structured than the soils under CM (WRa= 4.1±1.1) –in brackets, average and standard deviation, respectively– (Table 3) because the former holds the same relative fraction of water with more absolute energy in its porous system (Armindo and Wendroth, 2016). However, this difference between treatments was not statistically significant due to a large value of one the repetition on CM treatment (Table 3, in italics), though the rest of the values showed a relative low dispersion (see Table 3).

*Specific comments*. All the specific comments were considered and the text modified, accordingly. Note the reviewer that some specific comments related to crop management issues (i.e., cover crop, fertilization) were already treated (see above).

Finally, the English will be improved in the final draft after incorporating all the modifications and suggestions of all the reviewers.

**References**

[revised manuscript text omitted]

---

## Author Comment (AC8)

I strongly support field research and the cropping system approach instead of individual practice comparison and believe in the extra value of long-term experiments. Thus, I feel that this research will definitely contribute to the existing knowledge. Nevertheless, as the other reviewers also commented, the manuscript needs improvements before being published.

I considered all the comments of the other reviewers and will not repeat them as I mostly agree with both. I will indeed stress though that when in a cropping system the term sustainability is included then all aspects should be considered (social economic and environmental).

I hope that my recommendations along with those of the other reviewers will help you to improve the manuscript.

Thank you very much for your consideration and positive reaction. Your comments and suggestions will undoubtedly help to improve the manuscript.

The title would be more representative if it was something like "the effects of long-term conservation management on topsoil structure".

It is true that our findings are too narrow to be a comprehensive contribution to the sustainability of Mediterranean agrosystems considering all the aspect that sustainability development really involved. In fact, our main objective is the evaluation of an innovative -and pioneer in our region (Navarre)- soil and crops management strategy on topsoil properties. Moreover, there is -to our knowledge- no other agricultural field in the whole region of Navarre where soil and crop management as proposed herein (OPM) is practiced and even less for almost two decades, with the exception of precisely our small OPM test area. The uniqueness of our OPM area and then the relevance of our work would be better and clearly pointed out in the text, especially in the Introduction and in section 2.1. (Study site [former 'zone'] and treaments).

Following the reviewer suggestion and in accordance also to reviewer#1 a more appropriate title could be: *Effects of innovative long-term soil and crop management on topsoil properties of a Mediterranean soil based on detailed water retention curves.*

In the introduction, a more thorough review of the effects of the specific management practices on soil quality or at least on the WRC and soil aggregates should be included, and specific scientific questions and hypotheses should be structured.

We agree, the Introduction would be improved by incorporating the following information:

"Conservation agriculture (CA), and other soil management strategies implying a reduction of tillage have been reported to reduce soil degradation in different agroecological situations (Verhulst et al., 2010; Sartori et al., 2022), and in some cases are designed for this purpose (Virto et al., 2015).

The reasons reported for its adoption in Europe are several. In Northern Europe soil erosion control, soil crusting in loamy soils and the need to increase soil organic C storage, as well as soil trafficability are widely cited as reasons for CA implementation (Lahmar et al., 2007). In the Mediterranean countries, soil water storage and water-use efficiency can be added to this list of reasons (De Turdonnet et al., 2007). However, different studies show that the effectiveness of CA in solving these problems can be site-dependent (Costantini et al., 2020; Chenu et al., 2019). In fact, the most widely reported benefits of CA in Southwestern Europe in relation to erosion are the increased soil infiltrability and/or the protective effect of crop residues on the soil surface (Gómez et al., 2009; Espejo-Pérez et al., 2013; Virto et al., 2015), although this seems to be related to the type of soil and to the presence and activity of earthworms. In Spain, the soil water-retention capacity has been observed to be greater in semi-arid land under no-tillage (Fernández-Ugalde et al., 2009; Bescansa et al., 2006).

Other positive effects of CA on soil quality observed in semi-arid rainfed agricultural systems in Spain are related to soil organic C and nutrients storage (Ordóñez Fernández et al., 2007)".

On the other hand, we prefer not to include a hypothesis but instead the objective was reformulated more concisely: "The objective of this study was to assess the continuous application, throughout 18 years, of an innovative soil and crop management –in comparison with conventional management– for the improvement of the soil physical condition, and the optimization of the soil water balance, in rainfed cereal agrosystems in semi-arid land. This evaluation will be carried out through the analysis of detailed SWRCs and soil structure, i.e., the size-distribution of stable macro- and microaggregates and their relation to organic C storage."

The section materials and methods should be written in a clearer way.

Starting from the description of the treatments where more details should be included, such as the frequency and type of organic amendment and cover crops rather than occasional use. Also the type of mineral fertilization the quantities and frequency. Possibly a table with the crop rotation history and amendments application would be useful.

The reviewer is right. Description of the cover crops should be added to the text, and can be explained as follows:

"In OPM, both grain and straw were also removed in the 11 first years of implementation, and only stubble remained on the surface of soil when direct seeding was implemented with minimal soil perturbation. Since then, and for the 7 remaining years, the procedure was slightly modified, and only grain was removed at harvest. Therefore, chopped straw and stubble remained on the surface of the soil before direct seeding with no disruption of the soil surface. At the same time, cover crops were introduced in the system thought it is a risky practice in rainfed Mediterranean agrosystem characterized by warm and dry summers. As such, summer cover was routinely granted in this system by letting spontaneous vegetation grow in the summer, after harvest. This vegetation was dried with herbicides before seeding the crash crops in the Fall. Also, only one year the winter crop used was *Vicia villosa* Roth, and served as a cover crop for sorghum (*Sorghum vulgare* L.), which was successfully grown in the spring-fall season despite the limiting water availability in the area."

Also, more information about fertilization should be added to the text, and can be explained as follows:

"In both treatments, mineral fertilization consisted of phosphorus addition before seeding (120-150 kg·ha⁻¹ of triple superphosphate 0-46-0) and nitrogen supply of 180 kg N·ha⁻¹ (split and distributed into two cover dressings at 60 kg N·ha⁻¹ and 120 kg N·ha⁻¹ in January and March, respectively) as urea. Organic fertilization was not used in any of the study treatments until 2021, in which an organic amendment was applied to the soil without disturbing the surface in the OPM treatment. After harvest, pig slurry was applied with an average concentration of 2.5 kg N·m⁻³, by means of a tanker equipped with a system of hanging pipes that deposit the product a few centimeters above the ground and at a time close to a forecasted rainfall event. The application rate was 60 m³·ha⁻¹ of slurry. These rates are within the legal limits established by legislation for groundwater protection against pollution caused by nitrates from agricultural sources (EU Directive 91/676 (Council of the European Union, 2008)), as the area is within a vulnerable watershed according to this Directive."

The soil's initial characteristics (for the variables that are available) will give valuable information on the effects. That is an extra value of the long term experiments and you should present it.

Comparing the changes undergone by the soil under the different agricultural systems with the soil in its initial or natural state –a sort of control plot– would be ideal. However, soils in natural conditions are not found in the entire study area: they are all agricultural soils. Instead, we considered that our *initial* soil is that under conventional management (CM), and we intend to find out changes in its physical conditions induced by the proposed soil and crop management after a long period of time (almost two decades).

Focusing on the sampling procedures and especially the mixed subsample for the determination of the distribution of the aggregate, it is not clear how the required replication for the statistical analysis (at least 3 samples) happened as you indicate that "three sub-samples per sampling point, which were then mixed to obtain a composite sample)". As it is written it is not clear if you ended up with one composite sample per treatment or with three samples one at each sampling point.

There are 3 composite samples for treatment each of them consisted of three sub-samples. We will clearly indicate this in the final version.

In L: 105 you mention "Organic fertilization was not used in any of the study treatments until the experiment year in which an organic amendment was applied to the soil without disturbing the surface in the OPM treatment." What do you mean by experiment year? Before you mentioned that in the OPM there is frequent application of organic amendments. Do you mean before the beginning of the experiment? Ie 18 years ago?   Rephrase the sentence.

Sorry for this ambiguity, experiment year means 2021; this sentence should be reformulated: "Organic fertilization was not used in any of the study treatments until 2021, in which an organic amendment was applied to the soil without disturbing the surface in the OPM treatment".

In the results as well as in Table 1 you should indicate where the differences were significant. Which post-hoc test did you perform?

The reviewer is right. At the request also of reviewer #2, Table 1 has been completed (see below) and significant differences have been marked in bold. The statistical analysis used was the ANOVA test.

Table 1. Physical-chemical properties of the topsoil (0-30 cm) in OPM and CM treatments and the textural characterization of both treatments. Mean ± standard deviation of the mean (n = 3). Statistically significant differences ($p < 0.05$) are in bold.

| Treatment | Optimized (OPM) | Conventional (CM) |
|---|---|---|
| Bulk density (0-5 cm) (g·cm$^{-3}$) | 1.26 ± 0.05 | 1.26 ± 0.15 |
| pH | 8.00 ± 0.05 | 8.01 ± 0.01 |
| Organic C (%) | **1.80 ± 0.10** | **1.51 ± 0.14** |
| CE (µS·cm$^{-1}$) | **483 ± 5.66** | **795 ± 4.24** |
| Carbonates (%) | **31.6 ± 0.09** | **32.5 ± 0.14** |
| Sand (Coarse) (%) | 5.05 ± 0.08 | 5.79 ± 0.33 |
| Sand (Fine) (%) | 30.9 ± 1.00 | 31.7 ± 1.25 |
| Silt (%) | 47.2 ± 1.23 | 43.7 ± 0.93 |
| Clay (%) | **16.9 ± 0.46** | **18.5 ± 0.46** |
| **Texture (USDA)** | *Loam* | *Loam* |

In Figure 4, you should clearly mention what the error bars represent, you should include error bars in the in-between Magg fractions also to allow comparisons, and letters on top of each bar to indicate significant differences between the two treatments.

Thanks for your sound suggestion. Following the reviewer's recommendations, we would modify Figure 4 as follows, including all error bars, and a sentence has been added to the caption to indicate the significance of differences between treatments:

Figure 4. Size-distribution of stable aggregates and individual particles in the soil under OPM (a) and CM (b) Magg: Macroaggregates; magg: microaggregates; mMagg: microaggragates within macroaggregates. s+c: silt+clay fraction; cPOM: coarse particulate organic matter >250 μm and sand particles. The error bars represent the standard error, which is the standard deviation divided by the square root of the sample size. All aggregate fractions are significantly different ($p < 0.05$) between OPM and CM, with the exception of mMagg.

[Figure]

Focusing more on the aggregates the authors refer to the hierarchical aggregate order concept by Tisdall and Oades (1982). The wrongly present that the authors cited, mention that the magg are bonded together to form Magg. What is actually mentioned in the majority of the cited papers (eg Oades, 1984; Six et al., 1999, 2004;) is what was revised by Oades (1984) and validated by Angers (1997) that magg are formed within the Magg and not before these. The authors should correct all this paragraph and explain their results accordingly.

We agree with the reviewer on this correction, and have changed the text accordingly. In fact, our purpose with this part of the study was to test the effect of OPM on soil aggregation, and on the stabilizing agents of soil structure. Our findings that OPM resulted in more stable macroaggregates can be related to the observed gains in coarse particulate organic matter and microbial activity and biomass, which can be considered related to the stabilization of macroaggregates (inside which microaggregates would form).

In the results and discussion section, the authors should interpret their results in terms of causes, compared them with the existing literature in a more detailed way, and give practical recommendations according to their results.

In our original submission we had tried to interpret our results as suggested by the reviewer, but also considering the extension limitations of a manuscript of this type.

Following this comment and other from other reviewers, we envisage to extend this discussion at different levels:

- First, in relation to the SWRCs, we will complete the information with changes proposed by the reviewer #2 and #5. On the one hand, we will incorporate a new bimodal equation suggested by reviewer #2 and proposed by Dexter et al. (2008). Furthermore, following the recommendations of this same reviewer, the adjusted parameters of the van Genucthen equation will be added (see more details in the answers to reviewer #2).

  On the other hand, following the suggestion of reviewer #5 we have determined a new index based on the integration of SWRCs and named water retention energy index (WRa) (Armindo and Wendroth, 2016). The accuracy of this index depends on the degree of detail of the SWRCs. Additionally, this index presents an adequate sensitivity for smaller-scale, high-precision applications and for capturing the dynamic evolution of the soil physical state (Armindo and Wendroth, 2016) (see more details in the answers to reviewer #5).

- Second, in relation to aggregation and organic C, we aim to develop the discussion by adding a new discussion on the relevance of the hierarchical model of aggregation, and its relation to organic matter cycling, as follows:

  "The main implication of this hierarchy is that agricultural management primarily affects the less stable macroaggregates, while the more stable microaggregates are less influenced. Implicit in this concept is the fact that aggregates form sequentially (Jarvis, 2012).

  Although the relationship between organic matter cycling and soil structural stabilization has been observed to be soil-dependent (Rasmussen et al., 2018), and the calcareous nature of the studied soil may interact with it by stabilizing Magg and magg to a greater extent than in Ca-free soils (Fernández-Ugalde et al., 2011; Rowley et al., 2018, 2021), these results suggest that the response of soil structure to the reduction of tillage and the increase in organic C inputs corresponded to that observed previously in other soil types (Six et al., 2004; Fernández-Ugalde et al., 2016), and in soils of the same type in the region (Virto et al., 2007; Yagüe et al., 2016)."

- Third, in relation to organic C storage, as indicated in the response to reviewer #1 (Joan Bouma), we are proposing to add a whole new section to the R&D section entitled "Organic C storage and soil microbial diversity".

- Finally, in the conclusions of our work, we aim to provide some preliminary recommendations for agricultural soil management in the region, based on the design of our OPM system.

[revised manuscript text omitted]

---

## Author Comment (AC9)

I read with pleasure your MS entitled "Evaluation of a long-term optimized management strategy for the improvement of cultivated soils in rainfed cereal cropland based on water retention curves" by Aldaz-Lusarreta et al. The MS deals with the effect of an optimized agronomic protocol on water retention curve, pore size distribution (PSD) and aggregate stability. The Authors concluded that the soil under the optimized system reflected a better quality –or less degradation– compared to conventional management after 18 years of adoption. The MS falls within the Journal's scope and will interest Journal's readers however I have some major and minor issues to draw to your attention.

We appreciate your thorough review and your helpful comments and suggestions that significantly helped to improve our manuscript.

The structure of the abstract should be re-arranged by clearly dividing it into classical sections (introduction, M&M, results, conclusions). In particular, a few sentences regarding M&M should be added to introduce the reader to the work you did.

In fact, the abstract already comprises all these parts each of them as separated paragraphs, but they were automatically merged when uploaded. In addition, since the limited words normally allowed in the abstracts some sections had to be minimized. For instance, the introduction was practically reduced to the presentation of objectives. Anyway, if the Editor finally allows us to send a new draft incorporating all the modifications and suggestions from you and the rest of the reviewers, we will do our best to prepare a better and more complete abstract following your suggestions.

What is the caution of applying organic amendments under no-till? Nitrate Directive and, the more recent NEC Directive claim for greater caution in applying N sources. How can you meet environmental criteria by broadcasting organic waste on the soil surface and not incorporating it? This would result in greater ammonia volatilization and a high risk of leaching. Consequently, also the nitrogen use efficiency would result very low without improving the soil structure due to the low amendments-soil particle mixing.

In the OPM treatment, the measure adopted for the application of the organic amendment in the form of slurry was the distribution by means of a system of hanging tubes that deposit the product almost at ground level, and the moment of application close to a forecasted rainfall event, to minimize volatilization.

Thus, the paragraph has been reformulated to include more information on organic fertilization and its mode of use:

"In both treatments, mineral fertilization consisted of phosphorus addition before seeding (120-150 kg·ha-1 of triple superphosphate 0-46-0) and nitrogen supply of 180 kg N·ha-1 (split and distributed into two cover dressings at 60 kg N·ha$^{-1}$ and 120 kg N·ha$^{-1}$ in January and March, respectively) as urea. Organic fertilization was not used in any of the study treatments until 2021, in which an organic amendment was applied to the soil without disturbing the surface in the OPM treatment. After harvest, pig slurry was applied with an average concentration of 2.5 kg N·m$^{-3}$, by means of a tanker equipped with a system of hanging pipes that deposit the product a few centimeters above the ground and at a time close to a forecasted rainfall event. The application rate was 60 m$^3$·ha$^{-1}$ of slurry. These rates are within the legal limits established by legislation for groundwater protection against pollution caused by nitrates from agricultural sources (EU Directive 91/676 (Council of the European Union, 2008)), as the area is within a vulnerable watershed according to this Directive."

The Authors used an indirect method (capillary method) to estimate the PSD however nowadays a lot of direct non-destructive methods are available for a direct assessment of PSD. Therefore, the limitations of your study as related to the adopted methods should be included and discussed in detail.

The reviewer suggests a good point of discussion. In fact, there are several techniques for PSD characterization apart from SWRC (Wardak et al., 2022): (i) X-ray computed tomography (XRCT); (ii) Mercury intrusion porosimetry (MIP); (iii) Electricity resistivity tomography (ERT); and also micromorphological analysis (MA) (e.g., Pagliai et al., 2004).

The **XRCT** is a non-destructive imaging technique which allows the differentiation of materials based on their X-ray attenuation properties. This provides 3D information about soil porosity structure, but not about interconnection of pores.

The **MIP** is a technique used for the evaluation of porosity at micro and meso scale using a porosimeter. This employs a pressurized chamber to force mercury to intrude into the voids in a porous substrate. Although mercury moves only along interconnected pores, its high surface tension limits its access to larger pores connected by relatively small pores. This may lead to a somewhat mischaracterization of the pore network.

The **ERT** is a geophysical technique for imaging sub-surface structures from electrical resistivity measurements. This approach provides 3D information from soil regarding its porous architecture. For instance, unlike compacted soil layer, interconnected pore network may allow the conduction of water and thus increase resistivity. However, resistivity also correlates with many other soil properties (e.g., root systems, mineral surface conductivity) (Abidin et al., 2014; Cimpoiaşu et al., 2020) which constrains data interpretation.

The MA allows a 2D characterization of the pore space –e.g., pore size distribution, shape and pores orientation pattern– from undisturbed soil samples by means of electro-optical image-analysis. Like XRCT, MA does not provide information on pore interconnection, even less being just a 2D analysis.

If we compare all the above techniques with the present one (by using SWRCs), we can say that this approach does not provide information about the porous architecture but it enables an suitable characterization of those interconnected, *functional* pores, and more reliable than that from MIP since water is used instead of mercury. It should be noted that if the SWRC is subjected to several wetting and drying cycles the pore size distribution and other soil morphological properties can be affected – due to hysteresis phenomenon- (Pires et al., 2020). However, our SWRCs were not submitted to any wetting and drying cycle. In short, we understand that other methods are available, but we believe that soil water dynamics and porosity may be well evaluated through the analysis of SWRCs.

The authors presented the water retention curves but only tested the statistical differences between some relevant points (e.g., at saturation). Contrarily, a direct statistical comparison between the curves might give more value to your study.

Thanks for your sound suggestion. Taking advantage of our continuous SWRCs and following the reviewer suggestion we have determined the water retention energy index (WRa) (Eq.1) (Armindo and Wendroth, 2016) obtained from numerical integration including all the points of each SWRC. Needless to say, the accuracy of this index is highly conditioned by the degree of detail of the SWRCs.

$$WR_a = \int_{\theta_{pwp}}^{\theta_{fc}} h(\theta)\, d\vartheta \qquad\qquad (1)$$

Where $\Theta_{fc}$ and $\Theta_{pwp}$ is the volumetric water content at field capacity and permanent wilting point, respectively; $h$ is suction (kPa).

WRa quantifies the total absolute energy that has to be applied by the soil to hold water in its pores between field capacity ( $\Theta fc$) –i.e., after the water drainage process becomes negligible– and wilting point ( $\Theta pwp$) or any moisture point $\Theta j$, where $\Theta pwp \leq \Theta j < \Theta fc$.

This index presents an adequate sensitivity for smaller-scale, high-precision applications and for capturing the dynamic evolution of the soil physical state (Armindo and Wendroth, 2016). More precisely, in the case of two SWRCs measured before and after some natural or anthropogenic changes (e.g., tillage), these energy indices can be used to quantify the change in soil physical quality status (Armindo and Wendroth, 2016). For instance, Fuentes-Guevara et al. (2022) found significantly correlation between hydraulic-energy based indices –including WRa– with some physical properties before and after land leveling operations, indicating their capacity to capture soil structure changes.

We have determined the index for the suction range between field capacity (ca. 10 kPa) and a moisture content corresponding to ca.150 kPa (maximum operating value of the Hyprop device) (Table 1).

Table 1. Absolute water retention energy (WRa) values for the two treatments (OPM, CM).

| Treatment | WRa (kPa) |
|---|---|
| OPM - repetition 1 | 4.8 |
| OPM - repetition 2 | 4.9 |
| OPM - repetition 3 | 4.1 |
| CM - repetition 1 | *5.4* |
| CM - repetition 2 | 3.5 |
| CM - repetition 3 | 3.3 |

The soil under OPM (WRa= 4.6±0.5) seems to have better structured than the soils under CM (WRa= 4.1±1.1) –in brackets, average and standard deviation, respectively– (Table 1) because the former holds the same relative fraction of water with more absolute energy in its porous system (Armindo and Wendroth, 2016). However, this difference between treatments was not statistically significant due to a large value of one the repetition on CM treatment (Table 1, in italics), though the rest of the values showed a relative low dispersion (see Table 1).

LL85-86: please insert standard WRB soil classification

We have added to the proposed new version if the Editor allows for it to be submitted, the WRB soil classification as indicated by the reviewer:

"Agricultural fields under two contrasting agricultural management strategies on identical soil types (*Fluventic Haploxerepts* (Soil Survey Staff, 2014)) (*Fluvic Cambisol,* (WRB, 2015)), with loam texture at the upper soil horizon) and use (rainfed cereal cropping) were randomly selected in the municipality of Garinoain (Navarre) (42,59843° N, 1,64959° O)."

A more in deep discussion of your data as related to other published studies is missing in particular in the PSD section. There is a lot of literature on how different agronomic managements impact PSD. Just for example:

- Conservation Agriculture Had a Poor Impact on the Soil Porosity of Veneto Low-lying Plain Silty Soils after a 5-year Transition Period. Piccoli, I., Camarotto, C., Lazzaro, B., Furlan, L., Morari, F. Land Degradation and Development, 2017, 28(7), pp. 2039–2050.

- Zero tillage has important consequences for soil pore architecture and hydraulic transport: A review. Wardak, D.L.R., Padia, F.N., de Heer, M.I., Sturrock, C.J., Mooney, S.J. Geoderma, 2022. 422,115927

Thank you very much for the useful recommended papers. Other published studies will be incorporated into the discussion as suggested by the reviewer. The main statements to be added are summarized next:

"In works in which SWRs were used for the long-term study of pore size distribution in no-tillage (NT) and conventional tillage (CT) management, it was found that there is no unanimity in the results obtained (Wardak et al., 2022).

In the case of macropores, generally, soil macroporosity has been reported to increase in NT systems, especially in the surface soil layers (Wardak et al., 2022). However, there are studies that, like ours, identified a decrease in macropores in the NT treatment, particularly in the surface layer (0-20 cm) (Wardak et al., 2022). In a silty clay loam soil, Lipiec et al. (2006) observed that the pore system of the soil under CT presented greater macroporosity, with the differences between tillage treatments being more pronounced in the 0-10 cm depth than in the 10-20 cm depth. Similar results were obtained in clay soils by Tuzzin de Moraes et al. (2016) and Borges et al. (2019) who identified significantly higher macroporosity in the CT treatment compared to NT. The authors mention that this is because soil plowing in CM causes pore breakage, which contributes to an increase in macroporosity. In contrast, Imhoff et al. (2010) and Gao et al. (2019) saw increased macroporosity in the NT treatment in a silty loam soil and a sandy loam soil, respectively. This disparity of results in different studies suggest a site-dependency of this effect, which supports the need for site-specific assessments.

On the other hand, in the study of micro- and mesopores, most studies have observed an increase of these pores in NT treatment compared to CT. Examples are the works of Borges et al., (2019), Lipiec et al. (2006) and Tuzzin de Moraes et al. (2016) whose analysis of SWRCs showed a higher volume fraction of microsopores and mesopores under NT than under CT. However, in the study by Imhoff et al. (2010) they recorded a decrease in micro- and mesopore volume under NT treatment. Similarly, Gao et al. (2019) saw reduced mesoporosity in NT soil, observing no significant effect of such reduction on soil hydraulic properties."

The conclusion section needs to be re-arranged by removing the summary-like part (LL301-316) and should go beyond the study results.

If the Editor finally allows us to send a new draft incorporating all the modifications and suggestions from all the reviewers, we will do our best to prepare better and more concise conclusions. However, we prefer to follow a common practice among researchers when writing conclusions which is to include some piece of results before giving any real conclusion on this finding.

**References**

Abidin, M. H. Z., Saad, R., Ahmad, F., Wijeyesekera, D. C., and Baharuddin, M. F. T.: Correlation Analysis Between Field Electrical Resistivity Value (ERV) and Basic Geotechnical Properties (BGP), Soil Mech. Found. Eng., 51, 117–125, https://doi.org/10.1007/s11204-014-9264-x, 2014.

Armindo, R. A. and Wendroth, O.: Physical Soil Structure Evaluation based on Hydraulic Energy Functions, Soil Sci. Soc. Am. J., 80, 1167–1180, https://doi.org/10.2136/sssaj2016.03.0058, 2016.

Borges, J. A. R., Pires, L. F., Cássaro, F. A. M., Auler, A. C., Rosa, J. A., Heck, R. J., and Roque, W. L.: X-ray computed tomography for assessing the effect of tillage systems on topsoil morphological attributes, Soil Tillage Res., 189, 25–35, https://doi.org/10.1016/j.still.2018.12.019, 2019.

Cimpoiaşu, M. O., Kuras, O., Pridmore, T., and Mooney, S. J.: Potential of geoelectrical methods to monitor root zone processes and structure: A review, Geoderma, 365, https://doi.org/10.1016/j.geoderma.2020.114232, 2020.

Council of the European Union: Council Directive 91/676/EEC of 12 December 1991 concerning the protection of waters against pollution caused by nitrates from agricultural sources, , L 269, 1–15, 2008.

Fuentes-Guevara, M. D., Armindo, R. A., Timm, L. C., and Faria, L. C.: Examining the land leveling impacts on the physical quality of lowland soils in Southern Brazil, Soil Tillage Res., 215, 0–3, https://doi.org/10.1016/j.still.2021.105217, 2022.

Gao, L., Wang, B., Li, S., Wu, H., Wu, X., Liang, G., Gong, D., Zhang, X., Cai, D., and Degré, A.: Soil wet aggregate distribution and pore size distribution under different tillage systems after 16 years in the Loess Plateau of China, 173, 38–47, https://doi.org/10.1016/j.catena.2018.09.043, 2019.

Imhoff, S., Ghiberto, P. J., Grioni, A., and Gay, J. P.: Porosity characterization of Argiudolls under different management systems in the Argentine Flat Pampa, Geoderma, 158, 268–274, https://doi.org/10.1016/j.geoderma.2010.05.005, 2010.

Lipiec, J., Kuś, J., Słowińska-Jurkiewicz, A., and Nosalewicz, A.: Soil porosity and water infiltration as influenced by tillage methods, Soil Tillage Res., 89, 210–220, https://doi.org/10.1016/j.still.2005.07.012, 2006.

Pagliai, M., Vignozzi, N., and Pellegrini, S.: Soil structure and the effect of management practices, Soil Tillage Res., 79, 131–143, https://doi.org/10.1016/j.still.2004.07.002, 2004.

Pires, L. F., Auler, A. C., Roque, W. L., and Mooney, S. J.: X-ray microtomography analysis of soil pore structure dynamics under wetting and drying cycles, Geoderma, 362, 114103, https://doi.org/10.1016/j.geoderma.2019.114103, 2020.

Soil Survey Staff: Keys to soil taxonomy, USDA-Natural Resour. Conserv. Serv. Washington, DC., 12, 360, 2014.

Tuzzin de Moraes, M., Debiasi, H., Carlesso, R., Cezar Franchini, J., Rodrigues da Silva, V., and Bonini da Luz, F.: Soil physical quality on tillage and cropping systems after two decades in the subtropical region of Brazil, Soil Tillage Res., 155, 351–362, https://doi.org/10.1016/j.still.2015.07.015, 2016.

Wardak, D. L. R., Padia, F. N., de Heer, M. I., Sturrock, C. J., and Mooney, S. J.: Zero tillage has important consequences for soil pore architecture and hydraulic transport: A review, Geoderma, 422, 115927, https://doi.org/10.1016/j.geoderma.2022.115927, 2022.

WRB, I. W. G.: World Reference Base for Soil Resources 2014, update 2015. International soil classification system for naming soils and creating legends for soil maps., https://doi.org/10.1038/nnano.2009.216, 2015.

---

## Author Comment (AC10)

This article is an experimental study on the effect of conservation agricultural practices. Thus, an optimal treatment (OPM) is compared to a conventional treatment with a 15 cm ploughing. From my point of view the strong points of this study are: 1) the choice of treatments with the analysis of the condition obtained after 18 years of direct seeding. The documentation of long-term effects is a very valuable contribution. 2) the use of a structural stability test that I find relevant for this type of treatment. Around these strong points the study brings clear results, namely significant effects on macroporosity and structural stability.

However, the study also suffers from weak points. The experimental design remains minimalist (6 samples 3 per treatment). It is not clear whether the samples were taken on one field or whether 3 plots were observed with one sampling point per field. In the latter case, the lack of replication is a problem. Indeed, it is very likely that different fields, even if managed according to the same strategy, probably have different short and long term cropping histories. For soil characterisation, we remain somewhat in the midstream.

First of all, thank you very much for your consideration and positive reaction. Your comments and suggestions will undoubtedly help to improve the manuscript.

Indeed, the area of study including both CM and OPM tested points is relatively small (3 ha). It is important to note that there is to our knowledge, no other agricultural field in the whole region of Navarre where soil and crop management as proposed herein is practiced, and even less for almost two decades, with the exception of our relatively small area where OPM was introduced by a pioneer farmer nearly 20 years ago. The uniqueness of our OPM area and then the relevance of our work would be better and clearly pointed out in the text, especially in the Introduction and in section 2.1. (Study site [former 'zone'] and treatments).

There were 3 composite samples for each treatment (OPM and CM). Each of them consisted of three sub-samples randomly taken within the studied area.

Regarding the vague soil information, the reviewer is right, some of the soil information -clay, silt and sand contents- was originally included in the supplementary material of the manuscript. However, since this is a relevant information, it would be included in Table 1 along with the methods followed for the data determination. In addition, at the suggestion of reviewer #4, a statistical analysis of the parameters in Table 1 will be performed and those showing significant differences ($p < 0.05$) between treatments will be marked in bold.

Indeed, the functional consequences can only be partially addressed. On the SWRC, the water storage capacity, which likely is one of the most interesting property, cannot be addressed, as the measurements at the wilting point are not reported. Macroporosity is of interest for infiltration. Hydraulic conductivity measurements at saturation and/or near saturation would have allowed to interpret the clear differences in porosity and their consequence on the water flows. Previous studies have shown that the absence of tillage leads to a restructuring of the pore space which could lead to a continuity of macropores that might favour infiltraton. The link between structural stability and erosion risk, as mentioned in the conclusion, is also an interesting point that could be further investigated. To what extent can the difference in structural stabilities, as observed, play on the erosion risk.

Our analysis is focused on the suction range between 10 kPa (field capacity) and ca. 100 kPa, i.e., what would be the *available* –for use by plants/crops– water storage capacity. Besides, 150 kPa is the maximum operating suction value of the Hyprop device. However, gravimetric moisture content ($\Theta$) at 1500 kPa was also measured using a pressure plate: 10.4 % and 10.0 % for OPM and CM, respectively.

We agree that the better soil stability that the soil under OPM shows should play an important role in soil erosion mitigation. Although our findings do not allow us to go beyond this basic statement, we have developed this idea in the discussion of the future new version, under the light of recent literature on soil erosion control in Europe, as follows:

"Changes in soil erosion rates depend among others on climatic conditions, land use patterns, farmers' decisions and, agri-environmental policies (Panagos et al., 2021; Mosavi et al., 2020; Grillakis et al., 2020; Eekhout and De Vente, 2020; Paroissien et al., 2015)

Recent work has shown that studies associated with soil erosion are essential, both for agricultural conservation practices and to subsidize environmental planning, where economic practices must be calculated under conservationist principles (Panagos et al., 2021; Cruz et al., 2019; Mosavi et al., 2020; Plambeck, 2020). In particular, selective application of cover crops in soil erosion hotspots, combined with limited soil disturbance measures, has been recently recommended as an effective measure for partially or totally mitigate the effect of climate change on soil losses in Europe (Panagos et al., 2021)."

On the form the article could be improved.

1) The apriori statement that no tillage is optimal is questionable. The study should contribute to knowledge that could be used to support such an assertion.

The reviewer is right to avoid misinterpretation the title should be reformulated: *Effects of innovative long-term soil and crop management on topsoil properties of a Mediterranean soil based on detailed water retention curves.*

2) The state of the art in the introduction is very succinct. There is a lot of existing work that deals with the impact of conservation agriculture on soil properties. Some elements are given in the analysis of the results. A more extensive state of the art would allow to better situate the results. The subject is already well studied, but it is still important to document new cases (as that of the presented study) due to the complexity of the subject. Such a state-of-the-art analysis could highlight the originality of results obtained in the study.

Following the reviewer's advice, we have made a revision and have drafted the following paragraph to be included in the introduction:

"Conservation agriculture (CA), and other soil management strategies implying a reduction of tillage have been reported to reduce soil degradation in different agroecological situations (Verhulst et al., 2010; Sartori et al., 2022), and in some cases are designed for this purpose (Virto et al., 2015).

The reasons reported for its adoption in Europe are several. In Northern Europe soil erosion control, soil crusting in loamy soils and the need to increase soil organic C storage, as well as soil trafficability are widely cited as reasons for CA implementation (Lahmar et al., 2007). In the Mediterranean countries, soil water storage and water-use efficiency can be added to this list of reasons (De Turdonnet et al., 2007). However, different studies show that the effectiveness of CA in solving these problems can be site-dependent (Costantini et al., 2020; Chenu et al., 2019). In fact, the most widely reported benefits of CA in Southwestern Europe in relation to erosion are the increased soil infiltrability and/or the protective effect of crop residues on the soil surface (Gómez et al., 2009; Espejo-Pérez et al., 2013; Virto et al., 2015), although this seems to be related to the type of soil and to the presence and activity of earthworms. In Spain, the soil water-retention capacity has been observed to be greater in semi-arid land under no-tillage (Fernández-Ugalde et al., 2009; Bescansa et al., 2006).

Other positive effects of CA on soil quality observed in semi-arid rainfed agricultural systems in Spain are related to soil organic C and nutrients storage (Ordóñez Fernández et al., 2007)".

We believe it is appropriate to add the following comments about our soil and crop management proposal (OPM). As the reviewer is well aware, the crop performance under no-till is strongly dependent on the crop type and climate (Pires et al., 2013; Or et al., 2021) and also soil type. Then, no-till may not be suitable for all conditions (Pittelkow et al., 2015). In fact, conventional tillage -in carefully managed agricultural soils- may be imposed when no-tillage would lead to chronic and unacceptable yield losses (Or et al., 2021). Indeed, no-till often results in reduction in crop yields of ca. 10 % in some areas (Or et al., 2021). Although we do not have precise data on crop yields in both treatments (CM vs OPM) some rough data are available from the farmers managing the fields (see Table 1 below). From the data provided by the farmers, we can see that our proposed soil and crop management (OPM) is not inferior to the CM in terms of crop yields and then it seems to be indeed suitable for our local (Mediterranean) conditions.

Table 1. Average crop yield (2016-2021) of OPM and conventional agricultural fields under conventional tillage (CM), as reported by farmers.

| Crop | Yields (t/ha) | |
|---|---|---|
| | OPM | CM |
| Wheat | 6.8 - 9.3 | 5.5 - 7.0 |
| Barley | 5.8 - 8.0 | 5.0 - 6.5 |
| Rapessed | 2.0 - 4.0 | 2.0 - 3.0 |
| Legumes | 2.2 - 3.5 | 1.7 - 2.5 |

The uniqueness of our pioneer OPM area, and then the relevance of our work should be better and clearly pointed out in the text, especially in the Introduction.

3) The text is not always easy to read and follow, especially in the methodological and results analysis sections.

We would do our best to improve the readability of the MS after incorporating all the modifications and suggestions of all the reviewers.

**Specific points**

L105: when organic fertilizer was applied in OPM?

The text will be reformulated to include more information about organic fertilization:

"In both treatments, mineral fertilization consisted of phosphorus addition before seeding (120-150 kg·ha$^{-1}$ of triple superphosphate 0-46-0) and nitrogen supply of 180 kg N·ha$^{-1}$ (split and distributed into two cover dressings at 60 kg N·ha$^{-1}$ and 120 kg N·ha$^{-1}$ in January and March, respectively) as urea. Organic fertilization was not used in any of the study treatments until 2021, in which an organic amendment was applied to the soil without disturbing the surface in the OPM treatment. After harvest, pig slurry was applied with an average concentration of 2.5 kg N·m$^{-3}$, by means of a tanker equipped with a system of hanging pipes that deposit the product a few centimeters above the ground and at a time close to a forecasted rainfall event. The application rate was 60 m$^3$·ha$^{-1}$ of slurry. These rates are within the legal limits established by legislation for groundwater protection against pollution caused by nitrates from agricultural sources (EU Directive 91/676 (Council of the European Union, 2008)), as the area is within a vulnerable watershed according to this Directive."

Table 1: Soil granulometry would be a very valuable to interpret the results. This cannot be replaced by the fractions obtained for the structural stability.

The reviewer is right, granulometry –originally included in supplementary material– was added to Table 1.

L140: how the inflexion determined (second derivative = 0?).

Yes, the inflection point is the second derivative of the function

Section 2.4: as this method is the less common, it would interesting to introduce first the type of information that can be characterized and then give some key to understand the results. Can the stability properties be related to erosion risk?

In fact, the technique used is well documented in previous publications cited in the text (mainly Oliveira et al., 2019), so we prefer not to modify this section and to refer the reader to those publications for further details.

L195-198 it is a bit confusing to highlight that below field capacity the soil water capacity is comparable between treatments. This would mean that the water availability of the water storage (below field capacity) would not be affected by the treatments. It would be useful to analyse the moisture variations from field capacity to the driest points (idealy , the wilting point) that may lead to significant differences

As commented above we focused our analysis in the available water storage which is between around 10-100 kPa. However, the gravimetric water content at wilting point in both treatments – measured using a pressure plate– is roughly the same: 10.4 % and 10.0 % for the OPM and CM, respectively. We cannot go further in this analysis since there is no measurements points between 100-150 kPa and 1500 kPa (wilting point).

Table 2 and 3: the moisture at inflexion point differed a lot between treatments and analytic SWRC curve. Can you comment this.

In fact, the variation in moisture content corresponding to the infection point obtained with the van Genuchten equation (Table 2) is only around 2 %. This is true that this difference is higher for the sigmoidal equation adjusted to the experimental data (Table 3) because this approach presented a much higher dispersion both in S values and in the corresponding moisture content as pointed out in the text.

Section 3.3: how the porosity spectrum established?

As mentioned in the material and method the soil pores size-distribution was estimated from the equivalent radius obtained from the suction values of SWRCs by Jurin's law (1718) (equation 2).

L266-268: rephrase

L276-279: not clear

L280-283: rephrase

Sorry, the three sentences will be written more clearly.

L305-L308 a bit speculative without going further in the SWRC between Field capacity and the dry point the difference in water content is comparable

As commented above we focused our analysis in the available water storage which is between around 10-100 kPa.

L319-321: No data for infiltration. Results are over interpreted

This is true; it is rather speculative to directly relate infiltration to soil macroporosity. We will limit ourselves to the description of porosity, while indicating that it would be necessary to have infiltration measurements, ideally with controlled suction (e.g.,disc infiltrometer).

L322-L327 : establish clearly the criteria that support the judgement. For instance, macroporosity can be associated to potentially good infiltration which is in favour to CM. For me there is a bias in favour to OPM that start from the introduction and goes toward the conclusion. (I am not against OPM but in evaluating agricultural systems we must stay as objective as possible).

We agree, we will rephrase this paragraph as follows:

"In summary, our results suggest that this type of system that includes, among other techniques, the suppression of tillage, crop rotations, and the application of organic amendments, is providing good results from the viewpoint of the physical quality of the soil, and can be recommended for a higher soil sustainability in Mediterranean agrosystems. However, the optimized management analyzed herein is not currently a widespread practice in the region, most likely because the high initial investment and the farmer's concern that crop yield would be reduced. This work illustrates the need for an adequate assessment and dissemination to overcome these reluctances of farmers and other potential barriers to the adoption of this type of systems."

**In conclusion:** in its present form the paper cannot be published. I suggest major revision including, if possible, more results (texture, infiltration, wilting points…) and/or a clear articulation between the state of art and the findings.

**References**

[revised manuscript text omitted]

---

## Author Comment (AC11)

75: "….previously conventionally managemented" to "….managed"

Thank you for this correction.

80: "….a study is developed" – correct tense

Thank you for this correction.

85: A map showing the locations of the fields should help understand the results of this study. Hydrologic and topographic factors such as rainfall distribution, the slope of the land, etc. might have contributed to the differences in the SWRC. Also, the sizes of the fields and where the samples were taken are of great importance.

The answer to these observations can be found in the response sent to reviewers #2 and #3.

180: How many samples in total were collected and included in the statistical analysis per treatment (OPM or CM)? How many sites were included? I get the impression that only 3 samples were used per treatment. Depending on the size of the field, I do not think 3 samples are sufficient for a field-scale study considering soil's spatial heterogeneity.

The answers to these questions are in the response sent to reviewers #3 and #5.

225: "…..whichever method used for ist determination…" – correct "ist".

Thank you for the correction.

Figure 3. The legend should say OPM and CM.

You are right. We have modified the legend in Figure 4.

---

## Author Response (AR2)

Alaitz Aldaz-Lusarreta and co-authors

Universidad Pública de Navarra (UPNA)

Campus Arrosadia, 31006 Pamplona (Spain)

5                                                                                                    6 September 2022

Dear Dr Materechera,

We are very pleased by the good feedback of our manuscript. We have carefully responded in detail to the questions and concerns raised by the two anonymous reviewers in their corresponding response letters. Their insightful recommendations have been fully taken into account in the new revised manuscript.

10      The responses to each of the comments made by the two reviewers are detailed point by point below.

Sincerely,

Alaitz Aldaz-Lusarreta and co-authors.

15    Future studies should take a dynamic approach to soil water regimes applying widely available dynamic simulation models of the soil-water-atmosphere-plant system. Also soil sampling should extend over the entire rooting zone and not be limited to the topsoil.

Thank you very much for your helpful suggestions for future studies. In the conclusions of the revised manuscript, the need to take a dynamic approach to soil water regimes by using mathematical models is stated. As the need to expand the sampling

20    area to the entire rooting zone was already indicated in the previous version of the manuscript, we have included the need for dynamic approaches in the same paragraphs, which will read as follows in the new revised version:

"Further analysis at deeper soil layers –at least to the rooting depth– are necessary for a more complete assessment of the proposed optimized management. Moreover, to better understand changes in the soil hydrology, it is necessary to carry out experiments to determine infiltration rates, preferably under controlled suction. Finally, future studies should take a dynamic

25    approach to soil water regimes by taking advantage of the widely available dynamic simulation models of the soil-water-atmosphere-plant system."

30   The authors have made a great job revising the first version of the manuscript, which has been improved, and I appreciate their effort.

Many thanks to the reviewer for the good feedback on our manuscript and for these new insightful comments.

Nevertheless, there are still some aspects that demand a further revision:

1.   The description of the experimental site is better than the previous one. However, the use of more widely accepted climate

35   type such as Koppen-Geiger, (e.g. Peel et al. 2007), could be more helpful for the readers than the Papadakis scheme

mentioned                                    in                                    line                                    91.

I would not recommend the use of the Thornthwaite model for the estimation of the potential evapotranspiration, line 92,

since this model infer the net radiation from the air temperature what causes important deviations of the measured values,

in particular in semiarid areas (e.g. McMahon et al. 2013).

40   As suggested by the reviewer, in the revised manuscript the climate type and potential evapotranspiration are defined using the Koppen-Geiger and FAO Penman-Monteith models, respectively (lines 91-93):

"This is an area with a Csb type of climate according to the Koppen-Geiger classification (Gobierno de Navarra Meteorología y Climatología de Navarra, 2022; Peel et al., 2007). The mean annual reference evapotranspiration according to the FAO Penman-Monteith method is 1107 mm·year$^{-1}$. For crops in the rotation, the mean annual crop evapotranspiration is 326

45   mm·year$^{-1}$."

2.   The choice of the Water retention energy index of Armindo and Wendroth is correct. Nevertheless, the selection of the

values of integration limits in equation 4 is rather questionable. The estimation of the moisture content at the state of field

capacity, the upper limit of the integral, must be justified. The authors state that the moisture content at field capacity

correspond to the value at the inflection point of the soil water retention curve in lines 213-214. Why? There are several

50   proposals for the estimation of this moisture content but almost all of them have been justified. Therefore, the authors

must explain their decision, considering not the limitations of measuring method, (lines 204-206), but the physical

meaning of the limit in the water retention energy concept.

Similarly, the moisture content at the permanent wilting point needs another justification. As Czyz and Dexter (2012)

indicated, and experimental data confirm (e.g. Wiecheteck et al. 2020), the conventional permanent wilting limit, was

55   defined by the constraints of the usual measuring methods, not by the plant response. Again, the authors must properly

justify the selection of both limits of the water retention energy method.

We thank the reviewer for these detailed observations of our use of this index, as it has allowed to observe and improve some aspects in this part of the manuscript:

- Regarding the determination of the moisture content at field capacity, it corresponds in really to the value at the inflection point of the hydraulic conductivity vs soil water content curve and not to the inflection point of the SWRC, as ambiguously stated in the original manuscript. We thank the reviewer for finding this misleading ambiguity. We have corrected the error in the revised manuscript (line 220).

- The reviewer concern about the conventional wilting limit is relevant since it is true that the wilting point lacks a universal physiological –in terms of plant response– basis. We decided to use the classical concept of permanent wilting point at a suction of 1500 kPa as it is widely used in the literature, and then to facilitate comparisons with previous and future publications. However, as stated by Wiecheteck et al.'s (2020) when comparing the classical permanent wilting limit with the biological wilting of wheat and barley suggest that wilting depends on soil texture, with an occurrence of wilting at lower suction (i.e., wetter soil conditions) for sandy soils than for clay soils. This clarification was incorporated into the manuscript (line 160). In any case, a more in-depth discussion on the determination of more realistic values for this parameter is beyond the scope of this paper

3. The 'accumulated frequency' of pore size in line 314, caption, and ordinate-axis legend of Figure 3, is what in statistical terms is mentioned as probability distribution function of pore size.

Many thanks for the clarification. We corrected it in the revised manuscript (see Fig. 3 caption and ordinate-axis legend as well as line 318).

4. If the plot of the probability distribution function of Figure 3 uses a logarithmic abscissae-axis, why not change the suction-axis of Figure 2 from arithmetic to logarithmic?

In Figure 2, the limited range of suction values –from 0 to ~110 kPa– does not justify the use of a logarithmic scale, compared to the wide range of values in the abscissase-axis. Then, we prefer to keep the arithmetic scale in this axis in the figure.

5. I keep thinking that figure 1 is not necessary in the manuscript. A simple reference to the original one if Olivera et al. (2019) could be sufficient.

The reviewer is indeed right, we now realize that Figure 1 is rather unnecessary: we will therefore eliminate Figure 1 in the revised manuscript.

**References**

Gobierno de Navarra Meteorología y Climatología de Navarra: http://meteo.navarra.es/climatologia/selfichaclima.cfm?IDEstacion=81&tipo=MAN, last access: 30 August 2022.

90    Peel, M. C., Finlayson, B. L., and McMahon, T. A.: Updated world map of the Köppen-Geiger climate classification, Hydrol. Earth Syst. Sci., 11, 1633–1644, https://doi.org/10.5194/hess-11-1633-2007, 2007.

Wiecheteck, L. H., Giarola, N. F. B., de Lima, R. P., Tormena, C. A., Torres, L. C., and de Paula, A. L.: Comparing the classical permanent wilting point concept of soil (−15,000 hPa) to biological wilting of wheat and barley plants under contrasting soil textures, Agric. Water Manag., 230, 105965, https://doi.org/10.1016/j.agwat.2019.105965, 2020.

95

---

## Author Response (AR3)

Alaitz Aldaz-Lusarreta and co-authors

Universidad Pública de Navarra (UPNA)

Campus Arrosadia, 31006 Pamplona (Spain)

3 October 2022

Dear Dr Madejón,

Thank you very much for your comments. As indicated, we are uploading a new version of the text in which we have incorporated some changes following the last comments from the Topical Editor (Dr. Materechera). We thank Dr. Materechera

10 for his final thorough review and helpful comments and suggestions that helped to improve our manuscript. His insightful recommendations have been fully taken into account in the new revised manuscript.

The responses to each of the comments are detailed point by point below.

Sincerely,

Alaitz Aldaz-Lusarreta and co-authors.

15

**Topical Editor Revision**

**Comments to the author**:

Lines 95 and 99 -replace analysis with properties

Fine thanks. We replaced the term "analysis" with "properties".

20

Line 105 -instead of 'see below' provide specific reference section, table or figure.

Fine, instead of saying 'see below', we indicate the section: 'section 2.2'.

Line 106-it is not clear to say 'values were extrapolated to 30cm depth; what does this mean?

25   The Editor is right, this sentence is rather ambiguous. It was rephrased as follows: "However, based on field standard soil description, it is fairly safe to assume that the bulk density is roughly constant up to 30 cm depth".

Table 1 in the title, reference to topsoil as 0-30 cm is misleading, better remove 'topsoil'.

Fine, to avoid misleading the word 'topsoil' was replaced by just 'soil'.

30

Line 110-What is a subarea?

Subarea means a piece of the total study area.

Lines 110-112-What is the use of this? What do you mean by statistically different are in bold?; Why is the clay fraction in

35   bold?

This was made in response to reviewer #4, who said "In the results as well as in Table 1 you should indicate where the differences were significant." And our answer to his/her request was: "The reviewer is right […] Table 1 has been completed and significant differences ($p < 0.05$) have been marked in bold."

Line 126- What is meant by vegetation was 'dried with herbicide'?

40   The sentence was improved: "This vegetation was controlled with herbicides".

Line 125- Conventional tillage is up to 15 cm, so why do you use 30 cm as top soil?

In fact, for topsoil we mean 'upper soil layer', which is not necessary limited to the ploughing layer.

45

Line 137-Which directive is being referred to here?

Please, note that the reference to the Directive was already indicated at the end of the sentence i.e., EU Directive 91/676 (Council of the European Union, 2008).

50 Line 138- What is meant by 'it was checked'? please clarify.

The Editor is right, this sentence is not clear. It was reformulated as follows: "To avoid […] it was ensured that the two last crops of the rotation before the study both in OPM and CM were the same".

Line 141-Is sampling was carried out in fall and before soil preparation, where are these values? why were they collected at only 5 cm depth?

55 Samples were taken at 5 cm depth; however, as explained above (Line 25 of this response letter), these topsoil samples are indeed, representative of the soil up to 30 cm depth. This issue was indeed brought up by reviewer #1 (see comment and response #5 in the response letter to this reviewer), and clarified therein and in the text.

Line 148-Spell out SWRC at first mention.

60 Please note that SWRC was already spelt out at first mention in line 67.

Line 154-How did you determine the gravimetric water content?

Water loss and then gravimetric water content, was determined by weight differences of the soil sample register with the balance of the Hyprop© device (see line 154 of the manuscript, where it is said that both suction and weight were registered

65 in a continuous manner).

Lines 161-164: Did you measure the biological wilting point for comparison?

We only measured the classical permanent wilting point (i.e., at 1500 kPa) which facilitates comparison since it is a widely accepted threshold. However, we remind the reader that the classical wilting point must be taken with caution since it may

70 differ from the biological –hence, plant-dependent– wilting point as for instance in wheat and barley on sandy soils (Wiecheteck et al., 2020).

Lines 166-173: It is important to indicate here specifically which shape of the SWRC was used and why instead of just explaining the literature.

75 The different mathematical functions proposed in the literature to better adjust SWRCs depend on - or are conditioned by- the general shape of the curve, i.e. J-shape and S-shape. Then, we think it is important to show the corresponding references, before explaining our particular results (S-shaped curves, see lines 274-275 in the manuscript).

Lines 162-264: The use of three replicates for a study of this nature is of great concern as it comprises the rigour of the statistical analysis.

Please note the Editor that your concern regarding sampling was already addressed in the first revision in response to some reviewers. For instance, Reviewer #3 already pointed out this aspect in his revision.

We responded by saying that it was necessary first to clarify that "the OPM area is close to the CM area, and the whole surface accounts for only around three hectares. In relation to the comparability of the two practices, this means that fields are close enough to consider management as the only relevant factor of change. Three composite samples –including three sub-samples each– for each treatment (OPM and CM) were taken. Both composite samples and subsamples were  randomly taken within the whole studied area."

In short, the experimental area is small and homogeneous in terms of soil characteristics, so that the number of samples, although small, would be sufficient for an acceptable statistical analysis. In the first revised manuscript we already made a better description of the experimental area indicating without the ambiguities of the original manuscript its size and homogeneity.

Line 264-What is meant 'statistical treatments'?
Sorry, it is 'statistical analysis'.

Check out the one sentence paragraphs!
Fine, we reformulated the whole text in order to avoid the unnecessary one sentence paragraphs.

What is the value of presenting the individual replicates and not just the averages?
The purpose of showing individual values in Table 5 was to better highlight the dispersion of values. But now we realize that this is redundant since the dispersion is sufficiently shown by the standard deviation already indicated in the text, so table 5 has been discarded in the final version of the manuscript.

What is meant by repetition in Table 5?
Table 5 has been eliminated.

Figures have been presented separately from the text (at the end of text), why not do the same with tables?
Sorry we did not correctly interpret the Copernicus Publications Word template since no specific instructions are given with respect to tables. We will relocate all the tables at the end of the text.

110 Please check the references for conformity to the format-there are some that are not complying.

Fine thanks. We checked it.

Lines 158-159 is not clear; Please rephrase for clarity

Fine, the statement was rephrased and then improved.

115

**References**

Council of the European Union: Council Directive 91/676/EEC of 12 December 1991 concerning the protection of waters against pollution caused by nitrates from agricultural sources, , L 269, 1–15, 2008.

Wiecheteck, L. H., Giarola, N. F. B., de Lima, R. P., Tormena, C. A., Torres, L. C., and de Paula, A. L.: Comparing the
120    classical permanent wilting point concept of soil ($-15,000$ hPa) to biological wilting of wheat and barley plants under contrasting soil textures, Agric. Water Manag., 230, 105965, https://doi.org/10.1016/j.agwat.2019.105965, 2020.